



# Improved rain-rate and drop-size retrievals from airborne and spaceborne Doppler radar

Shannon L. Mason[1,2], J. Christine Chiu[1,2], Robin J. Hogan[1,3], and Lin Tian[4,5]

[1]Department of Meteorology, University of Reading, Reading, UK
[2]National Centre for Earth Observation, University of Reading, Reading, UK
[3]European Centre for Medium-range Weather Forecasts, Reading, UK
[4]NASA Goddard Space Flight Centre, Greenbelt MD, USA
[5]Morgan State University, Baltimore MD, USA

*Correspondence to:* Shannon Mason (s.l.mason@reading.ac.uk)

**Abstract.** Satellite radar remote-sensing of rain is important for quantifying of the global hydrological cycle, atmospheric energy budget, and many microphysical cloud and precipitation processes; however, radar estimates of rain rate are sensitive to assumptions about the raindrop size distribution. The upcoming EarthCARE satellite will feature a $94\,\text{GHz}$ Doppler radar alongside lidar and radiometer instruments, presenting opportunities for enhanced global retrievals of the rain drop size distribution.

In this paper we demonstrate the capability to retrieve both rain rate and a parameter of the rain drop size distribution from an airborne $94\,\text{GHz}$ Doppler radar using CAPTIVATE, the variational retrieval algorithm developed for EarthCARE radar–lidar synergy. For a range of rain regimes observed during the Tropical Composition, Cloud and Climate Coupling (TC4) field campaign in the eastern Pacific in 2007, we explore the contributions of Doppler velocity and path-integrated attenuation (PIA) to the retrievals, and evaluate the retrievals against independent measurements from a second, less attenuated, Doppler radar aboard the same aircraft. Retrieved drop number concentration varied over five orders of magnitude betweenlight rain from melting ice, and warm rain from liquid clouds. Doppler velocity can be used to estimate rain rate over land, and retrievals of rain rate and drop number concentration are possible in profiles of light rain over land; in moderate warm rain, drop number concentration can be retrieved without Doppler velocity. These results suggest that EarthCARE rain retrievals facilitated by Doppler radar will make substantial improvements to the global understanding of the interaction of clouds and precipitation.

## 1 Introduction

Satellite remote sensing of rain is important for quantifying the global water and energy cycles. Even light rain and drizzle make significant contributions to global precipitation (Berg et al., 2010; Haynes et al., 2009), while profiling measurements can be used to estimate the vertical transfer of latent heat (Nelson et al., 2016) and microphysical processes (Lebsock and L'Ecuyer, 2011; Wood et al., 2009). The intensity and drop size distribution (DSD) of rain are related to persistent errors in weather and climate models, which frequently produce excess drizzle from shallow maritime clouds (Stephens et al., 2010). Improved



instrumentation and retrieval algorithms for the satellite remote sensing of rain are therefore priorities for earth observation and model evaluation.

The first spaceborne cloud and precipitation radars facilitated significant advances in the detection and measurement of rain, especially over the oceans. The $14\,\mathrm{GHz}$ precipitation radar aboard the tropical rainfall measurement mission (TRMM; Kummerow et al., 1998) measured moderate and heavy precipitation in the tropics. The more sensitive $94\,\mathrm{GHz}$ cloud profiling radar aboard CloudSat (Stephens et al., 2002) is capable of measuring very frequent light rainfall not detected by TRMM, which amounts to $10\,\%$ of the total tropical maritime precipitation (Berg et al., 2010). CloudSat measurements of the profile of light rain have shown that around $70\,\%$ of marine precipitation falls as drizzle, $50-80\,\%$ of which evaporates before reaching the surface (Rapp et al., 2013). The high sensitivity of $94\,\mathrm{GHz}$ radar allows profiling measurements of light rain and drizzle, at the cost of significant radar attenuation in moderate to heavy rain.

To reliably estimate rain rates from profiles of radar reflectivity, the attenuation of the radar beam due to liquid hydrometeors must be quantified. The path-integrated attenuation (PIA) can be estimated from the reduction of the ocean surface backscatter relative to nearby clear-sky profiles (Meneghini et al., 1983). PIA estimated from this surface reference method is used in the rain retrieval algorithms of both TRMM (Iguchi et al., 2000; Meneghini and Liao, 2000) and CloudSat (L'Ecuyer and Stephens, 2002; Haynes et al., 2009) over the ocean; however, the surface backscatter over the land is much more variable and difficult to characterise. Consequently, operational CloudSat data products provide only rain detection over land, and not rain rate estimates (Haynes et al., 2009). Additional radar measurements that facilitate rain rate estimates over land would offer significant improvements over existing satellite capabilities.

Estimates of rain rate from limited radar measurements rely upon assumptions about the rain DSD. While the statistical properties of rain DSDs have been found to be broadly consistent and robust over time, whether measured in situ (Marshall and Palmer, 1948; Tokay and Short, 1996) or estimated by radar remote-sensing (Wilson et al., 1997; Illingworth and Blackman, 2002), the instantaneous microphysical properties of rain are observed to vary over many orders of magnitude (Testud et al., 2001). Assumptions about the drop number concentration in particular have been identified as a major source of uncertainty in TRMM and CloudSat estimates of rain rate (Iguchi et al., 2009; Lebsock and L'Ecuyer, 2011). To improve upon the uncertainties of satellite remote-sensed rain rate, there is a need for additional radar measurements with which to better characterise the rain DSD.

There are two approaches to improving rain retrievals with additional observations from satellite radars, both to assist in estimating rain rate over land, and to better constrain the rain DSD. The recent global precipitation measurement mission (GPM; Hou et al., 2014) uses the first dual-frequency radar in space to exploit differences in non-Rayleigh scattering at $35\,\mathrm{GHz}$ and $14\,\mathrm{GHz}$, to better constrain the rain DSD over land and ocean (Rose and Chandrasekar, 2006). Another approach is to use Doppler radar to measure the terminal fallspeed of raindrops, which is related to drop size. The Doppler spectrum has been used in ground-based radar retrievals to resolve vertical air motion (Atlas et al., 1973; Firda et al., 1999; O'Connor et al., 2005), distinguish cloud from precipitation (Frisch et al., 1995; Luke and Kollias, 2013), and to understand warm rain processes (Kollias et al., 2011b, a). Unfortunately in spaceborne radar applications the Doppler spectrum is broadened by the lateral motion of the radar platform with respect to the scattering hydrometeors (Illingworth et al., 2015), which distorts the higher



moments of the Doppler spectrum; consequently, only radar reflectivity, PIA, and mean Doppler velocity measurements are useful for spaceborne Doppler radar retrievals.

The EarthCARE satellite will feature a $94\,\mathrm{GHz}$ Doppler radar along with lidar and radiometers, for synergistic retrievals of clouds, aerosols and precipitation (Illingworth et al., 2015). In this study we develop a method for making an improved

estimate of rain rate by simultaneously retrieving information about the DSD from an airborne $94\,\mathrm{GHz}$ Doppler cloud radar. We use a variational retrieval methodology developed for radar–lidar synergy from EarthCARE (Illingworth et al., 2015), and exploit mean Doppler velocity to retrieve the raindrop number concentration. NASA's high-altitude ER-2 aircraft provides an ideal platform for satellite instruments and retrievals; we use ER-2 measurements from the Tropical Composition, Clouds and Climate Coupling field campaign (TC4) off Costa Rica and Panama in 2007 (Toon et al., 2010). A second $10\,\mathrm{GHz}$ Doppler

radar aboard ER-2 provides independent measurements at an unattenuated wavelength with which to evaluate the retrievals.

The structure of this paper is as follows. We first describe the aircraft measurements, the synergistic classification of hydrometeors, and the retrieval method (Section 2). The ambiguities of retrieving rain rate from attenuated radar profiles are discussed using synthetic measurements (Section 3), before $94\,\mathrm{GHz}$ radar retrievals of rain rate and drop number concentration are presented for three case studies, and the retrievals are evaluated against independent radar measurements at an unattenuated

wavelength (Section 4). We briefly consider applications of the dual-radar retrievals (Section 5), before summarising our key findings with a view to applications to EarthCARE retrievals (Section 6).

## 2    Data and retrieval methodology

### 2.1    Measurements used in the retrieval

The observations are from NASA's high altitude ER-2 aircraft during the TC4 experiment conducted over the tropical eastern

Pacific in July and August 2007 (Toon et al., 2010). ER-2 flies above the tropopause at an altitude of $20\,\mathrm{km}$ with a cruise speed around $200\,\mathrm{m\,s^{-1}}$. We analyse measurements from straight flight legs over the ocean, and average all measurements over $5\,\mathrm{s}$ intervals, so that each pixel of radar–lidar data has a $1\,\mathrm{km}$ along-track footprint.

The $94\,\mathrm{GHz}$ ($3.2\,\mathrm{mm}$ wavelength) cloud radar system (CRS; Li et al., 2004) and $9.6\,\mathrm{GHz}$ ($3.1\,\mathrm{cm}$ wavelength) ER-2 Doppler radar (EDOP; Heymsfield et al., 1996) measure radar reflectivity factor and mean Doppler velocity with a vertical gate spacing

of $37.5\,\mathrm{m}$. The $94\,\mathrm{GHz}$ radar reflectivity factor is calibrated against the $9.6\,\mathrm{GHz}$ radar near cloud-top (McGill, 2004), and the mean Doppler velocity measurements are calibrated using the surface signal (Li et al., 2004). The path-integrated attenuation (PIA) of the $94\,\mathrm{GHz}$ radar is estimated over the ocean using the surface reference technique (L'Ecuyer and Stephens, 2002; Haynes et al., 2007). In this study we focus on the retrieval of rain from the $94\,\mathrm{GHz}$ cloud radar, and use the $9.6\,\mathrm{GHz}$ radar for evaluation.

The cloud physics lidar (CPL; McGill et al., 2002) measures attenuated backscatter at $355\,\mathrm{nm}$, $532\,\mathrm{nm}$ and $1064\,\mathrm{nm}$, with linear polarisation ratio measured at the $1064\,\mathrm{nm}$ wavelength. In this study the $532\,\mathrm{nm}$ attenuated backscatter is used in the classification scheme to detect cloud top and to retrieve overlying ice cloud and liquid layers when the lidar signal is not fully attenuated.



The MODIS airborne simulator (MAS; King et al., 1996) and MODIS/ASTER airborne simulator (MASTER; Hook et al., 2001) imaging radiometers measure infrared (IR) and visible channels. Three visible channels are combined to create composite images of the case studies. Due to a failure in the MAS instrument, the MASTER instrument flew aboard ER-2 as a replacement after 29 July 2007 (Toon et al., 2010); the channels used in this study are common to both instruments.

Supplementary environmental data are required to complete the retrieval. Atmospheric temperature, humidity and ozone concentration are used to classify hydrometeor thermodynamic phase and estimate radar and lidar attenuation due to atmospheric gases. These variables are interpolated onto the flight track from the European Centre for Medium-Range Weather Forecasts (ECMWF) Interim Re-Analysis (ERA-Interim; Dee et al., 2011).

## 2.2 Target classification

Prior to the retrieval the contents of each pixel are classified based on a synthesis of radar and lidar measurements. We exploit the instruments' complementary sensitivities to different classes of hydrometeors to infer the presence of liquid cloud, rain and drizzle, and ice. This approach to radar–lidar target classification is similar to that described for CloudSat–CALIPSO in Ceccaldi et al. (2013), however the categories are simplified in this analysis.

A trade-off in radar and lidar remote sensing is that the hydrometeors with the strongest backscatter also strongly attenuate 15 the beam, weakening its penetration. The sensitivity of lidar to small ice crystals and cloud droplets makes it suited to detecting optically thin ice and liquid cloud, but lidar is therefore quickly attenuated in all but the optically thinnest clouds. In contrast, cloud radar is most sensitive to large hydrometeors such as ice aggregates and raindrops, and becomes fully attenuated in heavy rain. With the synergy of the two instruments we can use radar to detect optically thick clouds and light to moderate rain, while lidar detects optically thin ice and liquid cloud-tops missed by the radar.

The thermodynamic phase of targets is primarily determined by the atmospheric temperature from reanalysis, with further distinctions made using thresholds of radar and lidar measurements. At temperatures colder than $-40\,^{\circ}\mathrm{C}$ all targets are classified as ice, and at all temperatures warmer than $0\,^{\circ}\mathrm{C}$ as "warm" liquid cloud or precipitation. Rain and drizzle is inferred from radar reflectivities greater than $-15\,\mathrm{dBZ}$ (as in Haynes et al., 2011, and others), and may be colocated with warm liquid clouds detected by lidar. Between $-40\,^{\circ}\mathrm{C}$ and $0\,^{\circ}\mathrm{C}$ the thermodynamic phase of cloud water can be ice, supercooled liquid or, where 25 the two coexist, mixed-phase. First all targets detected by radar are classified as containing ice, due to the sensitivity of that instrument to the largest particles. Then liquid and ice as detected by lidar are distinguished based on the vertical gradient of lidar backscatter, which is higher in liquid cloud (Ceccaldi et al., 2013); this method of distinguishing liquid cloud is consistent with the method of Yoshida et al. (2010) using the lidar depolarization ratio. Where radar detects ice and lidar detects liquid, mixed-phase cloud is diagnosed.

The vertical structure and thermodynamic phase of clouds provide constraints on the retrieval of cloud and precipitation properties, but the entire profile is frequently not detectable by both instruments. Therefore the lidar is used to retrieve liquid clouds, but the presence of liquid droplets is an uncertainty in the classification scheme where only radar measurements are available. The lidar is included in the present work for its contribution to the classification of cloud through the vertical profile,





and for measuring the water content at cloud-top; however, the radar is the dominant instrument for the retrieval of rain. Finally, in profiles where the radar is fully attenuated by heavy rain, we assume that rain is continuous to the surface.

## 2.3 Retrieval methodology

Radar–lidar retrievals of profiles of rain and ice cloud are made using the Clouds Aerosols and Precipitation from mulTiple Instruments using a VAriational TEchnique (CAPTIVATE) algorithm, an earlier version of which was outlined in Illingworth et al. (2015). In this section we first describe the CAPTIVATE framework, then the main components pertinent to this study: the cost function, the state vector for rain, and the radar forward model. The retrieval is made by iteratively minimizing the cost function to find the state vector that corresponds to the smallest difference between observed and forward-modelled measurements. The state vector consists of the quantities or parameters of the rain DSD selected as retrieved variables. The forward models are used to estimate the measured variables given the state; the relevant measurements are radar reflectivity factor, PIA, and mean Doppler velocity. In this study we focus on the rain retrieval; details for other hydrometeors will be provided in subsequent papers.

### 2.3.1 Retrieval framework

The CAPTIVATE algorithm provides a framework for a variational, or optimal estimation, approach (Rodgers, 2000) to the inverse retrieval of the profiles of rain, ice and snow, liquid cloud and aerosols from one or more vertically-pointing active and passive instruments. CAPTIVATE is novel in that the measurements used and state variables retrieved are easily configurable, so that the same retrieval algorithm can be applied to spaceborne, airborne and ground-based applications. The treatment of retrieved state variables and the vertical representation of each class of hydrometeor can also be modified as appropriate. The variational approach allows for a robust treatment of uncertainties in the retrieval, subject to the appropriate selection of observational uncertainties, forward-model errors and physical constraints.

### 2.3.2 Cost function and minimization

The retrieval is made for each profile by iterating to find a state vector that minimizes a cost function, given by

$$J = \frac{1}{2}\delta\mathbf{y}^T\mathbf{R}^{-1}\delta\mathbf{y} + \frac{1}{2}\delta\mathbf{x}^T\mathbf{B}^{-1}\delta\mathbf{x} + J_c(\mathbf{x}) \tag{1}$$

where $\delta\mathbf{y} = \mathbf{y} - \mathbf{y}^f$ is the difference between the observed ($\mathbf{y}$) and forward-modelled ($\mathbf{y^f}$) measurements; $\mathbf{R}$ is the error covariance matrix of $\delta\mathbf{y}$, the sum of the error covariance matrices of the observations and the forward model; $\delta\mathbf{x} = \mathbf{x} - \mathbf{x}^a$ is the difference between the state ($\mathbf{x}$) and its a priori estimate ($\mathbf{x^a}$), and $\mathbf{B}$ is the error covariance matrix of the a priori; and $J_c(\mathbf{x})$ represents the physical and smoothing constraints on the state vector. The minimization of the cost function is carried out by iterating on the state vector beginning from the priors, in the direction of the first and second derivatives of the cost function (the Levenberg-Marquadt method; Rodgers, 2000).





### 2.3.3 Rain state variables

The rain DSD is given by a normalized Gamma function, of the form

$$N(D) = N_w \frac{\Gamma(4)}{3.67^4} \frac{(3.67+\mu)^{4+\mu}}{\Gamma(4+\mu)} \left(\frac{D}{D_0}\right)^\mu \exp\left(\frac{-(3.67+\mu)D}{D_0}\right). \tag{2}$$

This formulation is a function of three independent, physically meaningful parameters for the shape $\mu$, median drop size $D_0$, and drop number concentration $N_w$ of the DSD (Testud et al., 2001; Illingworth and Blackman, 2002). The shape factor $\mu$ is of secondary importance to $D_0$ and $N_w$ in terms of the radar reflectivity (Testud et al., 2001), and is poorly constrained by observations (e.g. Moisseev and Chandrasekar, 2007). In this retrieval we use $\mu = 5$, a value derived from both radar and distrometer studies (Wilson et al., 1997; Illingworth and Blackman, 2002). This simplifies the DSD to a 2-parameter function of $D_0$ and $N_w$. The uncertainty due to the assumption of fixed-$\mu$ DSD is estimated to be $\pm 15\%$ of the rain rate (Wilson et al., 1997), and is included in the uncertainty estimates of the retrieved quantities.

Our primary state variable is the rain rate,

$$R = \frac{\rho_w \pi}{6} \int\limits_0^\infty N(D) D^3 v(D)\, dD \quad [\mathrm{kg\,m^{-2}\,s^{-1}}], \tag{3}$$

from the third moment of the DSD where $\rho_w$ is the density of liquid water, $v(D)$ is the raindrop terminal velocity as a function of drop size from Beard (1976). Hereafter we scale $R$ by a factor of 3600 to express $R$ in units of $\mathrm{mm\,h^{-1}}$. For all retrievals a prior $R$ of $0.1\,\mathrm{mm\,h^{-1}}$ is used. The second state variable is $N_w$, so that one state variable is an integral over the DSD and the second is a parameter of the DSD. We take as the prior $N_w$ the number concentration intercept of the Marshall and Palmer (1948) DSD ($8 \times 10^6\,\mathrm{m^{-4}}$), and assume $N_w$ is constant with height in each profile.

When few observational variables are available, a single-parameter retrieval of $R$ can be made by holding $N_w$ constant, reducing the degrees of freedom so that $R$ is a function of $D_0$ alone. This is called the "$R$-only" retrieval, and is similar to CloudSat rain rate retrievals in which $N_w$ is assumed. When additional observational variables are available, such as the mean Doppler velocity, there may be sufficient information to also retrieve $N_w$; this is called the $R$-$N_w$ retrieval.

Table 1 summarises the rain state variables, their prior values and physical constraints applied to their representation in each vertical profile; for $R$-only retrievals the state vector $\mathbf{x}$ for a vertical profile with $n$ elements, or pixels, is given by

$$\mathbf{x} = \log\left[R_1 \cdots R_n\right]^{\mathrm{T}} \tag{4}$$

while for the $R$-$N_w$ retrieval the full state vector is

$$\mathbf{x} = \log\left[R_1 \cdots R_n \quad N_w\right]^{\mathrm{T}} \tag{5}$$

where a single $N_w$ is retrieved per profile. We use the natural logarithms of $R$ and $N_w$ as the state variables, to ensure that the values remain positive everywhere and that the algorithm converges faster. At moderate rain rates we expect $R$ to be close to invariant with height due to high fallspeeds and negligible evaporation (e.g. Matrosov, 2007); however, at low rain rates $R$ may





vary vertically, either due to evaporation in the lower atmosphere, or collision–coalescence processes in warm rain (Lebsock et al., 2011). Therefore to reduce noise and ensure smoothly varying rain rate with height, we retrieve each profile of $R$ as the coefficients of cubic spline basis functions.

**Table 1.** Rain state variables $\mathbf{x_i}$, their prior values $\mathbf{x_{ai}}$ and uncertainties $\sigma$. The profile of rain rate $R$ is always retrieved in each profile, while the drop number concentration parameter $N_w$ may be either retrieved or held at the prior; the melting layer thickness scaling $X_m$ is held constant in this study.

| $\mathbf{x}_i$ | $\mathbf{x}_{ai}$ | $\sigma(\log \mathbf{x}_i)$ | Vertical representation |
|---|---|---|---|
| $R$ | $0.1\,\mathrm{mm\,h}^{-1}$ | 3.0 | Retrieved as the coefficients of a cubic spline basis function. |
| $N_w$ | $8 \times 10^6\,\mathrm{m}^{-4}$ | 1.0 | Retrieved as constant in each profile ($R$-$N_w$ retrievals), or held at the prior value ($R$-only) |
| $X_m$ | 1.0 | 0.0 | Not retrieved. |

#### 2.3.4 Melting layer

We represent the melting layer by applying radar attenuation between the lowest pixel in each profile classified as ice, and the highest pixel classified as rain, provided the two pixels are contiguous. Following Matrosov et al. (2008), it is assumed that the two-way attenuation of the melting layer $A$ is proportional to the rain rate $R$ at the first pixel just below the melting layer, such that

$$A = kX_m R \quad [\mathrm{dB}] \tag{6}$$

where $k = 2.2\,\mathrm{dB\,km^{-1}(mm\,h^{-1})^{-1}}$ at $94\,\mathrm{GHz}$ (Matrosov et al., 2008). The thickness of the melting layer, and therefore the total attenuation, may also depend on the local temperature profile: as sufficient information to retrieve the total melting-layer attenuation may be available from the PIA and the attenuation inferred from the radar reflectivity gradient, we include the variable $X_m$ in the retrieval to represent the effect of melting layer thickness on radar attenuation; however, in this study the $X_m$ is held constant with a value of 1, allowing us to capture the effect of this uncertainty on the retrieved variables and their errors, without retrieving $X_m$.

### 2.4 Radar forward model

For a given state vector we estimate the corresponding measurements made by each instrument by forward modelling the scattering behaviour between the sensor and each gate, accounting for the effects of atmospheric gases, aerosols and hydrometeors.

The radar reflectivity factor of rain is a function of the sixth moment of the DSD,

$$Z = \int_0^\infty N(D)D^6 \gamma_f(D)\,dD \quad [\mathrm{dB\,Z}], \tag{7}$$




where $\gamma_f$ is the Mie–Rayleigh backscatter ratio at the radar frequency $f$, and is required for both $94\,\text{GHz}$ and $9.6\,\text{GHz}$ radars to account for non-Rayleigh scattering as drop sizes approach the radar wavelength.

All major scattering effects are modelled, so that the forward-modelled estimate of the apparent radar reflectivity is directly comparable to observations. Attenuation due to atmospheric gases and the dielectric factor of water are calculated from atmospheric temperature and humidity profiles. Multiple scattering effects for radar and lidar instruments can be estimated using Hogan (2008); however for the aircraft measurements used in this study we assume multiple scattering effects are negligible.

Radar attenuation due to hydrometeors is quantified at each gate by the extinction coefficient

$$k = \frac{\pi}{4} \int_0^\infty Q(D) N(D) D^2 \, dD \quad [\text{m}^{-1}] \tag{8}$$

where $Q(D)$ is the extinction efficiency calculated from Mie theory (Mie, 1908). The gradient of extinction can be related to the gradient of apparent radar reflectivity and used to estimate the rain rate as suggested by Matrosov (2007). A second approach to quantifying attenuation due to hydrometeors is to measure the two-way path-integrated attenuation

$$\text{PIA} = 2 \frac{10}{\log 10} \int_0^\infty k \, dz \quad [\text{dB}] \tag{9}$$

for each profile. PIA is estimated from the radar reflectivity at the ocean surface, and used as an observational measurement. Both approaches are implemented simultaneously; whereas the gradient method of Matrosov (2007) required an assumption of constant rain rate with height, within the CAPTIVATE variational scheme the gradient of $R$ and $k$ can be estimated simultaneously from the profile of radar reflectivity and PIA.

Finally the mean Doppler velocity is the reflectivity-weighted mean drop fallspeed,

$$\overline{v}_D = \frac{-\int_0^\infty N(D) D^6 v(D) \gamma_f(D) \, dD}{\int_0^\infty N(D) D^6 \gamma_f(D) \, dD} \quad [\text{m}\,\text{s}^{-1}] \tag{10}$$

where the terminal fallspeed of drops $v(D)$ is from the empirical formulation of Beard (1976) corrected for air density, and where positive velocities are toward the ground. The forward-modelled mean Doppler velocity is calculated assuming zero vertical air motion; therefore the difference between the forward-modelled and observed mean Doppler velocities will include a contribution from the vertical air motion, which is treated as an observational uncertainty.

The estimated uncertainties in the measurements are relaxed somewhat from the specified uncertainties for the CRS instrument (Li et al., 2004), so that the uncertainty in radar reflectivity is taken as $3\,\text{dBZ}$, and as $0.5\,\text{m}\,\text{s}^{-1}$ for mean Doppler velocity. The observed variables, their observational uncertainties and their vertical representation are summarized in Table 2.

## 3    Retrievals of rain rate with attenuated radar

The strong attenuation of $94\,\text{GHz}$ radar by liquid water, and especially by rain, presents a challenge for retrievals of rain from profiles of radar reflectivity (Hitschfeld and Bordan, 1954). For nadir-pointing radars the following ambiguity arises: when



**Table 2.** Observational variables $\mathbf{y_i}$ for Doppler radar, and their estimated uncertainties $\sigma(\mathbf{y_i})$ as used in the retrieval. Radar reflectivity $Z$ and mean Doppler velocity $\overline{v}_D$ are measured at each gate, while PIA is estimated from the radar reflectivity over the ocean surface.

| $\mathbf{y_i}$ | $\sigma(\mathbf{y_i})$ | Vertical representation |
|---|---|---|
| $Z$ | $3.0\,\mathrm{dBZ}$ | At each gate |
| $\overline{v}_D$ | $0.5\,\mathrm{m/s}$ | At each gate |
| PIA | $0.3\,\mathrm{dB}$ | Estimated in each profile from ocean surface reflectance |

the profile of apparent radar reflectivity decreases with range (toward the ground), the decrease could be due to either the attenuation of the radar beam, or to a physical change in the rain DSD (e.g., due to evaporation). These two possibilities each constitute a local minimum in the cost function, so that a profile of evaporating light rain with negligible attenuation may be wrongly identified as a profile of moderate rain with significant attenuation, and visa versa.

To illustrate the double-minimum problem, and to visualise how PIA and Doppler velocity may help resolve this retrieval ambiguity, we use the radar forward model to generate synthetic radar measurements assuming zero observational noise. In practice, measurement error and more complex profiles will introduce further uncertainties in the retrieval than in this synthetic case. Two synthetic profiles of rain are simulated, with constant rain rates of $0.05\,\mathrm{mm\,h^{-1}}$ and $5.0\,\mathrm{mm\,h^{-1}}$ below a level of $5\,\mathrm{km}$, and drop number concentration $N_w = 8 \times 10^6\,\mathrm{m^{-4}}$, representing a profile of light rain with negligible attenuation, and

of moderate rain with strong attenuation, respectively. In making the inverse retrieval of the profile of rain rate corresponding to a given profile of $94\,\mathrm{GHz}$ radar reflectivity, multiple solutions may be found depending on the prior rain rate: the low-$R$ and high-$R$ profile of $R$ (Fig. 1a) represent the two minima of the cost function for the retrieval from the radar reflectivity profile (Figs. 1b) corresponding to the $5.0\,\mathrm{mm\,h^{-1}}$ profile of rain (the "truth"). The radar reflectivity alone does not provide sufficient information to differentiate between the two solutions; however, the forward-modelled mean Doppler velocity profile (Fig. 1c)

and PIA (Fig. 1d) for the two solutions illustrate how additional observational variables may provide further information with which to resolve the ambiguity. The PIA differs by more than $30\,\mathrm{dB}$ between the two solutions, and is used effectively to differentiate light and moderate rain in CloudSat rain retrievals. The mean Doppler velocity profiles also differ significantly, with the "true" high-$R$ profile varying only slightly with altitude, while the gradient of mean Doppler velocity indicates a reduction of $D_0$ toward the surface in the low-$R$ profile. An additional advantage of the mean Doppler velocity is that it is not

affected by the partial attenuation of the radar.

We can quantify the contribution of the observational variables to resolving ambiguous retrievals using the cost function that is minimized within the variational retrieval scheme. A range of prior rain rates are taken as candidates for the starting-point of the retrieval, and for each prior $R$ the contribution of the observations to the cost function is calculated by

$$J_{\mathrm{obs}} = \frac{1}{2} \sum \frac{\left(\mathbf{y}^f - \mathbf{y}\right)^2}{\sigma_{\mathbf{y}}^2}. \tag{11}$$

which is equivalent to the first term of the cost function in equation (1). We can interpret the curve of $J_{\mathrm{obs}}$ (Fig. 1e) as showing the tendency of the retrieval algorithm to converge from any prior $R$ toward a local minimum in the cost function, wherein a





steeper curve indicates stronger convergence toward a more robust retrieval. To explore the contributions of the observational measurements, we run the retrievals for the two synthetic profiles with only radar reflectivity observations ("Z-only"), with one additional observational variable ("ZPIA", "Zv"), and with all available observations ("ZvPIA").

For the light rain profile, the cost function for the Z-only retrieval has a secondary minimum at $3.0-4.0\,\mathrm{mm\,h^{-1}}$. The bimodal shape of $J$ shows that the retrieval is sensitive to the choice of prior: if $R$ is less than $1.0\,\mathrm{mm\,h^{-1}}$, the retrieval will converge to the "true" $R$ profile, but if the prior $R$ is greater than $1.0\,\mathrm{mm\,h^{-1}}$ the retrieval will converge on the high-$R$ solution. Conversely, for the moderate rain profile, the Z-only retrieval will converge on a low-$R$ solution if the prior is less than around $0.5\,\mathrm{mm\,h^{-1}}$. These two solutions are compared in Figs. 1a–1d.

The effect of including PIA (dashed lines in Fig. 1e) is strongest for $R > 1.0\,\mathrm{mm\,h^{-1}}$, and this removes any sensitivity to the prior $R$, while the effect of including Doppler velocity (darker lines in Fig. 1e) is smoother across the full range of $R$ than that of PIA, and dominates at low $R$ where radar attenuation is negligible. When both PIA and Doppler measurements are used the effects are cumulative, and the gradient of $J$ shows even stronger convergence toward the unique solution.

This example provides a simple illustration of the bimodal cost function of an $R$-only rain retrieval with a strongly attenuating $94\,\mathrm{GHz}$ radar. Without additional observational measurements, a given profile of radar reflectivity may equally be explained by a strongly attenuating profile with constant $R$, or by a weakly attenuating profile in which $R$ decreases toward the surface. Either PIA or mean Doppler velocity are sufficient to resolve this ambiguity: PIA as a constraint on the total attenuation, and mean Doppler velocity on the profile of $D_0$. As PIA is typically estimated from the ocean surface backscatter, the availability of mean Doppler velocity to resolve these ambiguities presents an opportunity for Doppler radar estimates of rain rate over land.

## 4   Retrievals of rain rate and drop number concentration

We now put to use observations of both PIA and mean Doppler velocity, in addition to radar reflectivity, to make $R$-$N_w$ rain retrievals from $94\,\mathrm{GHz}$ Doppler radar measurements. Three cases of stratiform rain are selected from two ER-2 flights during TC4 (Fig. 2): two flight legs on 22 July 2007 include rain falling from melting ice, ranging in intensity from virga to heavy showers, and a case of light to moderate warm rain from liquid clouds is observed on 29 July 2007.

For each case the $R$-$N_w$ retrieval is performed using all available measurements from the $94\,\mathrm{GHz}$ radar: radar reflectivity, mean Doppler velocity and PIA. This "ZvPIA" retrieval is of primary interest for evaluating the full capabilities of the CAPTIVATE retrieval for a Doppler cloud radar; however, we are also interested in the capabilities of a retrieval when one of the observational measurements is not available, or has high observational uncertainty. When mean Doppler velocity measurements are not used (ZPIA), the observational variables are analogous to those available to CloudSat over ocean; however, unlike CloudSat rain retrievals, here we retrieve $N_w$ as well as $R$. Conversely, when PIA is not used (Zv) the observational variables are similar to those available to a Doppler radar over land, where the land surface cannot be sufficiently characterised to estimate PIA. The ZPIA and Zv retrievals of $R$-$N_w$ are less constrained by observations than the ZvPIA retrieval, and will therefore demonstrate some bimodal or poorly constrained retrievals similar to those demonstrated for $R$-only retrievals in





Section 3; nevertheless, we include ZPIA and Zv retrievals in order to demonstrate the information provided by the PIA and mean Doppler velocity separately, and to identify situations in which a satisfactory $R$-$N_w$ retrieval may be made with limited observational variables.

In each case the retrievals are evaluated by forward-modelling independent radar measurements at the unattenuated fre-
quency, and comparing the retrievals at a selected height above sea level.

### 4.1 Case 1: moderate rain from melting ice, 22 July 2007

Stratiform rain from melting ice provides a test of many of the simplifying assumptions made in rain retrievals. At moderate and heavy rain rates we expect $R$ to be close to constant with height, unless significant evaporation is evident (Haynes et al., 2009). $N_w$ may be expected to be close to values deemed typical by Marshall and Palmer (1948) or Testud et al. (2001), i.e.
between $2.0 \times 10^6 - 8.0 \times 10^6 \, \mathrm{m}^{-4}$, and constant with height (Tokay and Short, 1996). From in situ measurements of stratiform rain we expect median drop sizes in this case to be in the range $1.0 - 1.5 \, \mathrm{mm}$ (Tokay and Short, 1996).

Between 15:54 and 16:03 UTC on 22 July 2007 ER-2 overflew approximately $110 \, \mathrm{km}$ of stratiform precipitation around $50 \, \mathrm{km}$ south of the coast of Panama (Fig. 2). Radar, lidar and radiometer measurements (Fig. 3) reveal distinct regimes of light, moderate and heavy rain below a melting layer around $4 \, \mathrm{km}$ above sea level, contiguous with ice clouds with tops between
$6 - 10 \, \mathrm{km}$. The scene is overlain by cirrus between $10 - 15 \, \mathrm{km}$. In light rain between 15:54 and 15:55 UTC the $94 \, \mathrm{GHz}$ radar is barely attenuated. Moderate stratiform rain follows from 15:55 and 16:03 UTC, with a strong $9.6 \, \mathrm{GHz}$ bright band, and PIA between $5$ and $20 \, \mathrm{dB}$. Finally a heavy shower is embedded within the moderate rain between 16:01 and 16:02 UTC. In the latter regime the $94 \, \mathrm{GHz}$ radar is completely attenuated such that PIA saturates around $65 \, \mathrm{dB}$; $94 \, \mathrm{GHz}$ radar reflectivity and mean Doppler velocity measurements are therefore not available within the heaviest rain profiles.

The retrieved variables (Figs. 4a–4e), and forward-modelled $94 \, \mathrm{GHz}$ and $9.6 \, \mathrm{GHz}$ radar measurements (Figs. 4f–4j) are compared for the ZvPIA, Zv and ZPIA retrievals. We evaluate the retrievals at a height of $3 \, \mathrm{km}$ above sea level, approximately $1 \, \mathrm{km}$ below the melting layer.

#### 4.1.1 Moderate rain (15:55–16:01 and 16:02–16:03 UTC)

In the moderate rain regime the ZvPIA retrieval estimates rain rates of $1.0 - 2.0 \, \mathrm{mm\,h}^{-1}$ at the melting layer. In profiles with
strong attenuation (PIA up to $20 \, \mathrm{dB}$), $R$ is constant from the melting layer to the surface; conversely, in less attenuated profiles (with PIA around $10 \, \mathrm{dB}$) evaporation is evident, with $R$ reducing to $0.1 - 1.0 \, \mathrm{mm\,h}^{-1}$ at the surface (Fig. 4a). Estimates of $N_w$ are consistently between $10^6 - 10^7 \, \mathrm{m}^{-4}$ in this regime (Fig. 4c), close to the Marshall and Palmer (1948) value. $D_0$ is around $1.0 \, \mathrm{mm}$ at the melting layer, reducing somewhat toward the surface in profiles where evaporation is strong (Fig. 4d). Forward-modelled $94 \, \mathrm{GHz}$ radar measurements agree with observations at $3 \, \mathrm{km}$ (Figs. 4f–h), as expected since the retrieval
minimizes differences between the observed and forward-modelled variables. Comparison of the independent $9.6 \, \mathrm{GHz}$ radar measurements with those forward-modelled from the retrieved state shows very close agreement overall (Figs. 4i and 4j), although $9.6 \, \mathrm{GHz}$ radar reflectivity is overestimated by as much as $3 \, \mathrm{dB}$ in profiles with strong evaporation, especially between





15:58 and 16:00 UTC. Moderate rain profiles with significant evaporation are well-represented by the ZvPIA retrieval, however these profiles correspond to the strongest differences compared to the independent $9.6\,\mathrm{GHz}$ radar reflectivity.

We use the ZPIA and Zv retrievals of $R\text{-}N_w$ to demonstrate the contributions of mean Doppler velocity and PIA to a successful retrieval, and the ambiguities that arise in under-constrained retrievals. Both ZPIA and Zv retrievals are considerably more sensitive to the selection of priors than the ZvPIA retrieval. At $3\,\mathrm{km}$ above sea level, ZPIA estimates of $R$ are close to those of ZvPIA, but $N_w$ and $D_0$ differ, with ZPIA estimating a much higher concentration of smaller drops than ZvPIA. Forward-modelled mean Doppler velocity shows that this retrieval leads to significant errors of drop fallspeeds. The Zv retrieval tends to underestimate the rain rate in this regime by up to an order of magnitude, tending toward the prior $R$ of $0.1\,\mathrm{mm\,h^{-1}}$: while $D_0$ is well constrained by the mean Doppler velocity, without a constraint on the total attenuation from PIA, the retrieved drop number concentration is low; the forward-modelled observations confirms that the Zv retrieval meets reflectivity and mean Doppler velocity constraints, but tends toward representing weakly-attenuating profiles of rain; the forward-modelled $9.6\,\mathrm{GHz}$ measurements show that this retrieval leads to a significant under-estimate in radar reflectivity at the unattenuated wavelength.

Selected profiles of retrieved and forward-modelled variables at 15:58:00 UTC (Fig. 5) illustrate how the Zv and ZPIA retrievals differ from ZvPIA through the vertical profile. With constraints on PIA, both ZPIA and ZvPIA retrievals estimate profiles of $R$ that are close to constant with height; without a Doppler constraint on drop size, the ZPIA retrieval represents the observed PIA with a higher number of smaller drops than the ZvPIA retrieval, leading to a mean Doppler velocity around $1.0\,\mathrm{m\,s^{-1}}$ lower than observed. Conversely, the Zv retrieval has a strong constraint on drop size and therefore forward-modelled mean Doppler velocity, but without a constraint on PIA a low concentration of drops is estimated, with $R$ and $D_0$ decreasing toward the surface.

### 4.1.2 Light rain (15:54–15:55 UTC)

In the light rain regime, ZvPIA estimates $R$ in the range $0.002-0.1\,\mathrm{mm\,h^{-1}}$ and $N_w$ in the range $10^5-10^6\,\mathrm{m^{-4}}$. The lower rain rate corresponds to an observed $1.0\,\mathrm{m\,s^{-1}}$ decrease in $94\,\mathrm{GHz}$ mean Doppler velocity compared to the moderate rain regime; the retrieval resolves smaller drops in the light rain, with $D_0$ around $0.5\,\mathrm{mm}$. The forward-modelled $9.6\,\mathrm{GHz}$ radar measurements from the ZvPIA retrieval are consistent with independent observations.

Zv retrieves $R$ consistent with ZvPIA throughout the light rain regime, while ZPIA somewhat overestimates $R$ in these profiles. PIA is negligible and therefore provides little additional information in this regime: therefore the ZPIA retrieval represents a higher concentration of smaller drops as the retrieved $N_w$ tends to the prior. This sensitivity to the prior when observational information is limited was demonstrated in Section 3 and, as in that synthetic case, the ZPIA retrieval here could be improved with a more appropriate prior. In contrast, with Doppler velocity information to constraint drop size and PIA negligible, the DSD retrieved by Zv is very close to that of ZvPIA. The strong performance of Zv in light rain suggests strong potential for using Doppler radar for $R\text{-}N_w$ retrievals of light rain over land.



### 4.1.3 Heavy shower (16:01–16:02 UTC)

The upper limit of the $94\,\mathrm{GHz}$ radar frequency for rain retrievals is reached in the heavy shower, where PIA is saturated and no radar reflectivity or Doppler information is available below the melting layer. Based on the saturated PIA and the gradient of radar reflectivity in the first few gates, both ZvPIA and ZPIA retrievals estimate $R$ up to $10\,\mathrm{mm\,h^{-1}}$; large uncertainties in $R$ reflect the dearth of information available for retrievals in these profiles, however the $9.6\,\mathrm{GHz}$ radar measurements are broadly consistent with profiles in which ZPIA and ZvPIA estimate rain rates around $10\,\mathrm{mm\,h^{-1}}$. Without PIA information, the Zv retrieval interprets the deficit in radar reflectivity as a drop in rain rate and drop size, and adds significant uncertainty to the retrieved quantities. The estimates of $N_w$ vary over many orders of magnitude and are clearly unconstrained by observations in this regime; the $R$-$N_w$ retrieval is not warranted without sufficient observational information, however PIA does appear to be sufficient for to estimate $R$, with increased uncertainty.

In this case of tropical stratiform rain the $94\,\mathrm{GHz}$ radar is fully attenuated by rain rates around $10\,\mathrm{mm\,h^{-1}}$ falling from a melting layer around $4.0\,\mathrm{km}$ above sea level. In the midlatitudes, however, where melting layers are much shallower, successful $R$-$N_w$ retrievals should be possible up to higher rain rates before the radar is fully attenuated.

### 4.1.4 Joint frequencies of retrieved and forward-modelled variables

A more comprehensive evaluation of the retrievals against independent $9.6\,\mathrm{GHz}$ radar measurements can be made using the joint frequencies of retrieved state variables (Figs. 6a–6c) and forward-modelled $9.6\,\mathrm{GHz}$ radar measurements (Figs. 6d–6f) for each retrieval. The major modes in the rain retrieval are evident in the distributions of $R$ and $N_w$ relative to the priors (dashed lines), and in the distribution of $9.6\,\mathrm{GHz}$ radar reflectivity and mean Doppler velocity compared against observations (black contours). In the $9.6\,\mathrm{GHz}$ radar variables the moderate rain regime exhibits radar reflectivity around $20\,\mathrm{dBZ}$ and mean Doppler velocity between $6-7\,\mathrm{m\,s^{-1}}$, while the light rain regime has radar reflectivity between $0-5\,\mathrm{dBZ}$ and mean Doppler velocity around $3\,\mathrm{m\,s^{-1}}$.

The ZPIA retrieval has a dominant mode corresponding to the moderate rain regime, with $R$ between $0.5-2.0\,\mathrm{mm\,h^{-1}}$ and a higher $N_w$ with respect to the prior; without a constraint on mean Doppler velocity this retrieval represents a relatively high concentration of small drops. The corresponding forward-modelled measurements shows the small drop size leads to a significant underestimate of both mean Doppler velocity and radar reflectivity at $9.6\,\mathrm{GHz}$.

Without PIA information to constraint the total attenuation in the profile, Zv retrievals in the moderate rain regime tend toward weakly attenuated profiles with $N_w$ less than $10^6\,\mathrm{m^{-4}}$, where $R$ tends toward the prior. This lead to underestimates of radar reflectivity by more than $10\,\mathrm{dB}$; the mean Doppler velocity is reasonably well-constrained, but broadly underestimated by around $1\,\mathrm{m\,s^{-1}}$. Light rain profiles are represented with $N_w \approx 10^6\,\mathrm{m^{-4}}$, somewhat overestimating mean Doppler velocity.

ZvPIA resolves distinct modes for light and moderate rain regimes in the retrieved variables, and each mode corresponds well to the observed $9.6\,\mathrm{GHz}$ radar measurements: the moderate rain regime is represented with heavier rain than the Zv retrieval, but with a lower concentration of smaller drops than the ZPIA retrieval; the light rain regime is similar to that of the Zv retrieval, where the negligible PIA provides little additional information. Both rain regimes have $N_w$ around $10^6\,\mathrm{m^{-4}}$, consistent with





the average value of $2 \times 10^6 \, \mathrm{m}^{-4}$ for stratiform rain found by Testud et al. (2001). The $9.6 \, \mathrm{GHz}$ radar reflectivity is well-represented across both rain regimes, however the mean Doppler velocity shows that drop fallspeed is slightly under-estimated in moderate rain, and over-estimated in the light rain; this may be due to the retrieval assuming $N_w$ is constant with height in each profile, such that any variations in the DSD with height are expressed as changes in drop size, rather than in drop number concentration.

In summary, we have retrieved $R$ and $N_w$ of stratiform rain from melting ice across rain rates varying from light rain as low as $10^{-3} \, \mathrm{mm \, h^{-1}}$, to a heavy shower with $R$ up to $10 \, \mathrm{mm \, h^{-1}}$. The retrieved $N_w$ was around $10^6 \, \mathrm{m}^{-4}$ throughout the case, which is consistent with expectations for average drop number concentrations in this context; the exception is in the heavy rain shower where the $94 \, \mathrm{GHz}$ radar becomes fully attenuated, and insufficient information is available for a $R$-$N_w$ retrieval. The Zv retrieval, an analogue for Doppler radar retrievals over land, performed very well in light rain, but tended towards the $R$ prior in moderate rain profiles where attenuation makes profiles of radar reflectivity ambiguous. In this case retrievals of $R$-$N_w$ without Doppler velocity information tend to estimate $R$ broadly accurately, but retrieve a high number concentration of small drops, leading to errors with respect to the independent radar measurements; indeed, since the estimated $N_w$ were close to expectations in this context, a good non-Doppler retrieval of $R$ would be made by assuming the value of $N_w$.

## 4.2 Case 2: evaporating rain from melting ice, 22 July 2007

We now evaluate the $R$-$N_w$ retrieval for a case of very light rain falling from ice, much of which evaporates before reaching the ground. ER-2 overflew a $60 \, \mathrm{km}$ section of ice cloud $300 \, \mathrm{km}$ south of Costa Rica, between 13:12:00–13:17:30 UTC on 22 July 2007. Light rain was observed below ice clouds with tops between $10 - 12 \, \mathrm{km}$ (Fig. 7). Below the melting layer both $94 \, \mathrm{GHz}$ and $9.6 \, \mathrm{GHz}$ radar reflectivity are less than $10 \, \mathrm{dBZ}$ and decreasing toward the surface; the exception is a region of higher $9.6 \, \mathrm{GHz}$ radar reflectivity between 13:16:00–13:17:00 UTC, and $95 \, \mathrm{GHz}$ PIA is small but non-negligible, around $3 \, \mathrm{dB}$. In combination with low $94 \, \mathrm{GHz}$ PIA, the observations suggest stratiform light rain evaporating in the lower atmosphere, including significant virga.

Time series of retrieved variables (Figs. 8a and 8e), and forward-modelled $94 \, \mathrm{GHz}$ and $9.6 \, \mathrm{GHz}$ radar measurements (Figs. 8f and 8j) are evaluated against observations. We compare ZPIA, Zv and ZvPIA retrievals at a height of $4 \, \mathrm{km}$ above sea level, which is just below the melting layer.

ZvPIA makes a consistent retrieval of evaporating light stratiform rain, with $R$ between $0.1 - 0.2 \, \mathrm{mm \, h^{-1}}$ at the melting layer, down to a minimum detectable rate of $10^{-3} \, \mathrm{mm \, h^{-1}}$ near the surface or at the limits of the virga. In the heaviest rain profiles between 13:16–13:17, $R$ is around $0.1 \, \mathrm{mm \, h^{-1}}$ at the surface, with $D_0$ as large as $1.5 \, \mathrm{mm}$. Retrieved $N_w$ is consistently around $10^5 \, \mathrm{m}^{-4}$, an order of magnitude lower than the previous case of stratiform rain from melting ice, and significantly lower than the prior. Forward-modelled $9.6 \, \mathrm{GHz}$ radar variables shows very good agreement with independent measurements, which can be expected as there should be little ambiguity in the retrieval due to attenuation.



Similar to the light rain profiles of Case 1, both ZPIA and Zv retrievals make estimates of $R$ close to the ZvPIA retrieval. ZPIA retrievals slightly overestimates $R$ with $N_w$ again close to the prior, which is 2 to 3 orders of magnitude higher than ZvPIA estimates; the corresponding low $D_0$ of around $0.5\,\mathrm{mm}$ leads to significant differences between forward-modelled and observed mean Doppler velocities. Zv estimates of $D_0$ are well-constrained by mean Doppler velocity, and where PIA is

negligible the Zv retrieval is identical to that of ZvPIA. As noted in the previous case, this indicates that it should be possible to make an accurate $R$-$N_w$ retrieval of light rain over land with Doppler radar.

### 4.3 Case 3: warm rain, 29 July 2007

Another challenge for rain retrievals is to represent collision–coalescence drop growth processes in warm clouds. In warm rain we expect a higher concentration of smaller drops, with drops growing between cloud-top and the surface (Lebsock et al.,

10   2011).

On 29 July 2007 ER-2 overflew a $120\,\mathrm{km}$ section of precipitating warm marine cloud around $500\,\mathrm{km}$ south of Costa Rica between 12:41–12:51 UTC (Fig. 9). In the first part of the flight (12:41–12:45 UTC) observations suggest moderate rainfall corresponding to deeper cloud tops around $3.5\,\mathrm{km}$: PIA varies between $10-50\,\mathrm{dB}$ in narrow features, where $9.6\,\mathrm{GHz}$ radar reflectivity exceeds $20\,\mathrm{dBZ}$. The following section (12:45–12:51 UTC) is characterised by shallower stratiform cloud with tops

around $3\,\mathrm{km}$, and is associated with patchy light precipitation with PIA between $0-10\,\mathrm{dB}$.

Concurrent to the rain retrieval shown here, we use the lidar to retrieve liquid cloud, which also contributes to the attenuation of $94\,\mathrm{GHz}$ radar. The retrieved properties of the liquid cloud do not vary between the different retrievals compared here, and we do not evaluate the retrieval of cloud liquid water content in this study; however we acknowledge that the simultaneous retrieval of cloud and precipitation in warm clouds from $94\,\mathrm{GHz}$ radar is a source of uncertainty that warrants further consideration (e.g.

Haynes et al., 2009; Hawkness-Smith, 2010; Mace et al., 2016).

The retrieved variables (Fig.10a–10e) and forward-modelled radar measurements (Fig.10f–10j) are compared at $1\,\mathrm{km}$ above sea level, and compared against $94\,\mathrm{GHz}$ and $9.6\,\mathrm{GHz}$ radar measurements. We compare ZPIA, Zv and ZvPIA retrievals as in the previous cases. Warm rain or drizzle forming from collision and coalescence in liquid clouds can be easily distinguished from rain falling below ice clouds within the target classification scheme, so that physically appropriate choices for the priors

and the physical representations of state variables can be configured in CAPTIVATE for distinct warm and "cold" rain regimes; however, in this study we use the same prior $R$ and $N_w$ throughout.

#### 4.3.1 Moderate rain (12:41–12:45 UTC)

The ZvPIA retrieval resolves a strong increase of rain rate from cloud-top, where $R$ is between $0.1-0.5\,\mathrm{mm\,h^{-1}}$, to the surface, where $R$ increases to $1.0-10.0\,\mathrm{mm\,h^{-1}}$. Retrieved $N_w$ is consistently around $10^{10}\,\mathrm{m^{-4}}$ in the moderate rain regime, several

orders of magnitude greater than retrieved for rain from melting ice; the corresponding drop sizes are much smaller, with $D_0$ increasing from $0.1-0.3\,\mathrm{mm}$ at cloud-top, to $0.2-0.5\,\mathrm{mm}$ near the surface. The $94\,\mathrm{GHz}$ radar measurements correspond very well to the forward-modelled variables, and the $9.6\,\mathrm{GHz}$ radar reflectivity is also exceptionally close to the forward-model;





this is because at the small drop sizes in this context scattering is in the Rayleigh regime for both radar frequencies. The forward-modelled mean Doppler velocity at $9.6\,\mathrm{GHz}$ also tracks well with observations.

At $1\,\mathrm{km}$ the ZPIA retrieval strongly resembles ZvPIA; this includes matching estimates of $D_0$, despite having no constraint on drop size from mean Doppler velocity. The forward-modelled $94\,\mathrm{GHz}$ and $9.6\,\mathrm{GHz}$ mean Doppler velocities for the ZPIA retrieval are very close to observations. In contrast Zv also retrieves $D_0$ correctly, but frequently under-estimates $N_w$ by as much as 2 orders of magnitude: the Zv-retrieved DSD has fewer drops and negligible PIA at $94\,\mathrm{GHz}$, which corresponds to very large errors in forward-modelled $9.6\,\mathrm{GHz}$ radar reflectivity. Unlike the stratiform rain cases, here PIA is more important for an accurate retrieval than mean Doppler velocity: the mean Doppler velocity is less sensitive to the changes in terminal fallspeed due to variations in the sizes of small drops, while PIA in combination with radar reflectivity provide an effective constraint on the number concentration because only a DSD with many small drops satisfies the observed strong attenuation and low radar reflectivity.

### 4.3.2 Light rain (12:45–12:51 UTC)

In the light warm rain ZvPIA estimates patchy precipitation features with $R$ between $0.01 - 0.5\,\mathrm{mm\,h^{-1}}$ and median drop sizes around $0.1 - 0.3\,\mathrm{mm\,h^{-1}}$, similar to values at the tops of the deeper warm clouds, but without significant drop growth toward the surface. The retrieved $N_w$ in the lightest rain profiles is around $10^8 - 10^9\,\mathrm{m^{-4}}$, but returns to $10^{10}\,\mathrm{m^{-4}}$ where heavier rain features are evident. The forward-modelled radar reflectivities are close to observations at $1\,\mathrm{km}$, while the mean Doppler velocity again matches the lower range of measurements, but not the noisy peaks. The forward-modelled PIA is $0\,\mathrm{dB}$ in the lightest rain profiles, while measurements indicate non-zero attenuation of around $1 - 3\,\mathrm{dB}$. ZPIA estimates $R$ similar to ZvPIA in the stratus regime, but without a Doppler velocity constraint in the lightest profiles, ZPIA retrieves fewer, larger drops, with $N_w$ tending toward the prior at $8 \times 10^6\,\mathrm{m^{-4}}$. In contrast, Zv is very similar to ZvPIA in the light rain.

In summary, it is possible to retrieve $N_w$ across a range of $R$ from very light drizzle, up to $10\,\mathrm{mm\,h^{-1}}$ in the heaviest profiles of warm rain from liquid clouds. The contribution of PIA and mean Doppler velocity to $R$-$N_w$ retrievals in warm rain differs from that in rain from melting ice: while Doppler is required to retrieve $N_w$ when attenuation is low, it is possible to retrieve $N_w$ without Doppler in strongly attenuated profiles of warm cloud, where the combination of low radar reflectivity and high attenuation can only be due to a high concentration of small drops.

## 5 Dual-frequency radar retrievals

ER-2 aircraft measurements from TC4 provide a rare opportunity for airborne observations with multiple Doppler radars. In this study we have primarily used the $9.6\,\mathrm{GHz}$ radar to evaluate retrievals made with the $94\,\mathrm{GHz}$ radar; however, we can also use the dual-frequency radar measurements to exploit the different scattering behaviours and retrieve additional information about the DSD. Dual-frequency ratio (DFR) and differential Doppler velocity (DDV) techniques have been applied to retrievals



from ER-2 measurements during the CRYSTAL-FACE field experiment over Florida in 2002 (Liao et al., 2008, 2009), and Tian et al. (2007) exploited dual-frequency Doppler radars to retrieve rain DSD and vertical air motion for light stratiform rain from the same experiment. The CAPTIVATE framework can combine information from two radars, resolving differential non-Rayleigh scattering and mean Doppler velocities from multiple wavelengths.

The dual-frequency radar retrievals, with and without mean Doppler velocity, are compared against the $94\,\mathrm{GHz}$ retrieval for Case 1 on 22 July 2007, which covered a wide range of intensities, including a region in which the $94\,\mathrm{GHz}$ radar was fully attenuated (Fig. 11). The dual-frequency radar retrieval estimates of $R$ are consistent with those from $94\,\mathrm{GHz}$ retrievals, with the exception of the non-Doppler dual-frequency radar retrieval in the light rain, where a high concentration of small drops is estimated, leading to an over-estimate of $R$; in much of the lightest rain profiles the hydrometeors may be below

the sensitivity of the $9.6\,\mathrm{GHz}$ instrument, so that the dual-frequency radar retrieval tends toward that of the ZPIA retrieval from the $94\,\mathrm{GHz}$ radar. In the heavy shower, where the ZvPIA estimates of $Z$ have large uncertainties and $N_w$ is very poorly constrained, the dual-frequency radar retrievals use the unattenuated $9.6\,\mathrm{GHz}$ to estimate $R$ around $10\,\mathrm{mm\,h^{-1}}$; $N_w$ remains in the range $10^6 - 10^7\,\mathrm{m^{-4}}$ as in the surrounding moderate rain, and $D_0$ between $1.0 - 2.0\,\mathrm{mm}$. This confirms that the $94\,\mathrm{GHz}$ retrieval was capable of a cautious estimate of $R$ based on the gradient of radar reflectivity and saturated PIA; however, ZvPIA

estimates of $N_w$ cannot by justified when the radar is fully attenuated. The greatest errors in the non-Doppler dual-frequency radar retrieval are in forward-modelled Doppler velocity for the evaporating moderate rain profiles between 15:58–16:00 UTC, where a higher concentration of smaller drops is retrieved; in this circumstance the addition of Doppler velocity information leads to a stronger retrieval than a second radar wavelength. Overall the close agreement of the $94\,\mathrm{GHz}$ Doppler radar retrievals with the dual-frequency Doppler retrieval is a promising result, indicating that a single frequency Doppler radar is sufficient

for a retrieval of $R$ and $N_w$ within the limits of radar attenuation.

## 6    Discussion and conclusions

The upcoming ESA/JAXA EarthCARE satellite will include the $94\,\mathrm{GHz}$ cloud profiling radar, the first Doppler radar in space. In this study we have used aircraft measurements to investigate the prospects for improved global rain retrievals from a space-borne Doppler radar, with a focus on how Doppler velocity measurements stand to improve upon $94\,\mathrm{GHz}$ radar rain retrievals

from CloudSat in two key respects: (1) to facilitate rain rate estimates over land, and; (2) to reduce uncertainties in rain rate estimates by retrieving an additional parameter of the raindrop size distribution (DSD). Retrievals over a range of stratiform rain regimes were made using measurements from the $94\,\mathrm{GHz}$ Doppler radar aboard the ER-2 aircraft during the TC4 field campaign over the tropical Pacific in 2007, and evaluated against independent measurements from a second Doppler radar at an unattenuated wavelength.

The CAPTIVATE algorithm has been developed for rain, cloud and aerosols retrievals from the synergy of active and passive instruments; within the variational scheme, multiple observational variables can be combined as available, and the retrieved variables and their physical representation can be configured at runtime. It is therefore possible with CAPTIVATE to combine the information from multiple radar measurements, and to estimate uncertainties in retrieved variables propagated from uncer-





tainties in the observations and forward-models. We have shown that the ambiguities of rain rate retrievals from attenuated radar can be resolved by either PIA or mean Doppler velocity measurements, with the latter having potential applications to making estimates of rain rate over land with a Doppler radar; furthermore, information from both PIA and mean Doppler velocity can be used to retrieve the rain rate as a function of drop number concentration $N_w$, improving upon significant uncertainties in

rain rate estimates owing to assumptions about the DSD.

Retrievals of rain rate $R$ and drop number concentration $N_w$ using combined radar reflectivity, PIA and mean Doppler velocity from the $94\,\mathrm{GHz}$ radar were evaluated over three cases of tropical stratiform rain over the ocean. The selected cases covered a range of rain rates from virga to heavy showers, and rain from both melting ice and liquid clouds. The $94\,\mathrm{GHz}$ radar was fully attenuated by rain rates around $10\,\mathrm{mm\,h^{-1}}$ falling below a melting layer above $4\,\mathrm{km}$, and in slightly heavier rain

rates from liquid clouds with tops around $3\,\mathrm{km}$. The attenuation of the $94\,\mathrm{GHz}$ radar places an upper limit on the rain profiles that can be retrieved; however, in the mid-latitudes where the melting layer is lower, it may be possible to make retrieval up to higher rain rates before the radar is fully attenuated. The $9.6\,\mathrm{GHz}$ radar wavelength was subsequently included to make a dual-frequency radar retrieval (Section 5). In profiles where the $94\,\mathrm{GHz}$ was fully attenuated, the dual-frequency estimates of rain rate were consistent with those derived from the gradient of $94\,\mathrm{GHz}$ radar reflectivity. Retrieved values of $N_w$ ranged from

$10^5\,\mathrm{m^{-4}}$ in light rain from melting ice, with drop size $D_0$ around $1.0-1.5\,\mathrm{mm}$, up to $N_w$ of $10^{10}\,\mathrm{m^{-4}}$ in moderate rain from liquid cloud, where $D_0$ was around $0.1-0.3\,\mathrm{mm}$.

Mean Doppler velocity provides a strong constraint on $D_0$ that is not compromised by partial attenuation of the radar beam. In combination with radar reflectivity, mean Doppler velocity is sufficient to resolve the ambiguities of radar reflectivity profiles due to attenuation, supporting retrievals of rain rate over land where PIA cannot be estimated from the surface reference

technique. Furthermore, in combination with PIA, the Doppler velocity information provided sufficient information to make robust retrievals of $R$-$N_w$ across a range of rain regimes. In light rain with negligible PIA, Doppler velocity is sufficient to retrieve $R$ and $N_w$, suggesting the possibility for Doppler radar $R$-$N_w$ retrievals of light rain over land; however, in moderate rain rates PIA is necessary to constrain the retrieval. Satisfactory retrievals of rain rate may be made over land by assuming $N_w$, or PIA could be estimated from the land surface with a large observational uncertainty (as in Iguchi et al., 2009), which

may provide sufficient information to resolve the ambiguity between weakly and strongly attenuating profiles.

With a constraint on PIA, the gradient of radar reflectivity can be used to estimate $R$. While Doppler velocity is generally required to retrieve $N_w$, in moderate warm rain from liquid clouds the combination of low radar reflectivity and strong attenuation was sufficient to retrieve the high concentration of small drops typical of warm rain, without the need for Doppler velocity information: this finding may be applicable to retrievals of the drop number concentration in warm rain observed by CloudSat.

With the first Doppler radar in space, EarthCARE will make a significant contribution to the satellite remote sensing of rain. Mean Doppler velocity provides an effective constraint on drop size that can be exploited to estimate rain rate over land, and to reduce uncertainty in rain rate estimates through the vertical profile by retrieving the drop number concentration $N_w$ of the rain DSD, which must otherwise be assumed. As part of EarthCARE radar–lidar–radiometer synergy retrievals, improved global estimates of rain rate and drop size from Doppler radar will facilitate new insights into the interactions of clouds, aerosols and

precipitation through the atmospheric profile, and the global energy and water cycles.





*Acknowledgements.* This work was supported by the National Centre for Earth Observation (NCEO) and European Space Agency Grant 4000112030/15/NL/CT, with computing resources provided by the University of Reading. ER-2 data from TC4 are provided by NASA; we thank Dennis Hlavka (NASA-GSFC) for assistance with CPL data, and Stephen Platnick and Howard L Tan (NASA-JPL) with MAS/MASTER radiometer data. ERA-Interim data are produced and distributed by ECMWF, and hosted by the Centre for Environmental Data Analysis.





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





**Figure 1.** Above: Profiles of (a) retrieved rain rate, and (b) forward-modelled $94\,\text{GHz}$ radar reflectivity, (c) mean Doppler velocity and (d) PIA, for the two solutions to the retrieval from a synthetic profile; dashed lines show the values corresponding to the "true" profile of constant $R = 5.0\,\text{mm}\,\text{h}^{-1}$.

Below: (e) the observational component of the cost function ($J_{\text{obs}}$) for retrievals of two constant rain profiles with $R = 0.05\,\text{mm}\,\text{h}^{-1}$ and $R = 5.0\,\text{mm}\,\text{h}^{-1}$, initialised from a range of $R$ priors. Bimodal or ambiguous retrievals are evident when using radar reflectivity alone (Z-only; light solid lines), and compared against retrievals using additional observational variables (Zv, ZPIA, and ZvPIA; dashed and dark lines) to resolve the ambiguity.



**Figure 2.** Flight tracks of the NASA ER-2 high-altitude aircraft over the tropical eastern Pacific on 22 July and 29 July 2007 during the TC4 field campaign. The flight legs selected for case studies of stratiform rain are highlighted.





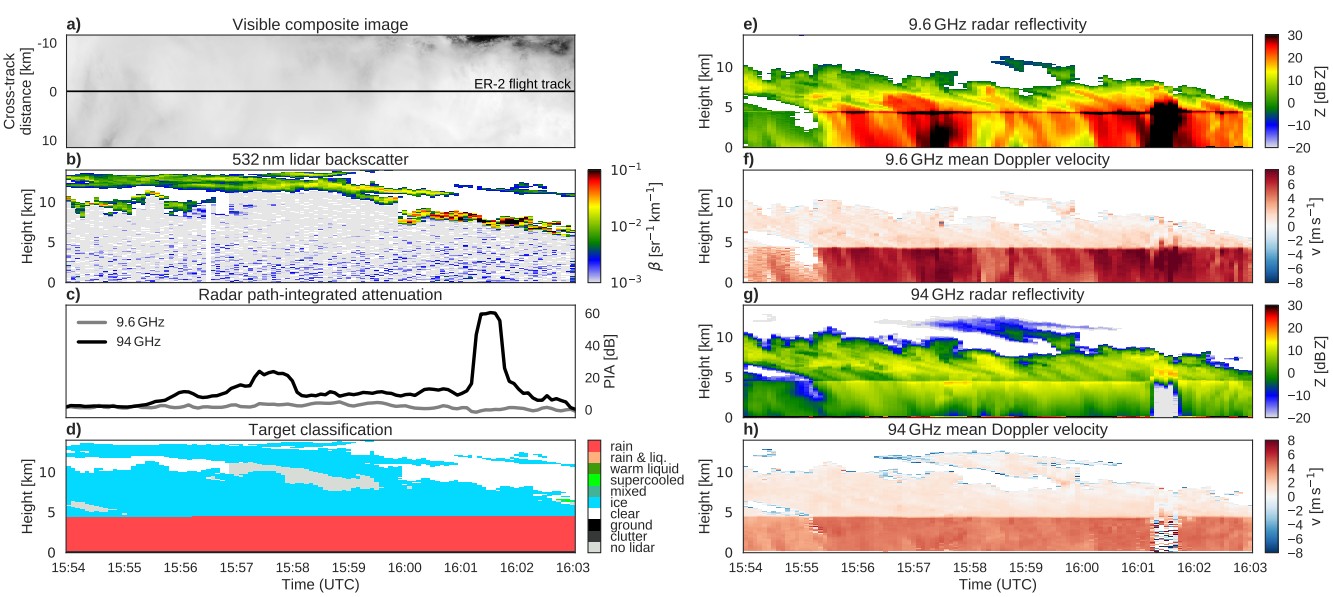

**Figure 3.** Selected measurements made by ER-2 instruments for Case 1 between 15:53 and 16:03 UTC on 22 July 2007 as part of TC4. Composite cloud scene (a) from MAS/MASTER visible channels, with the ER-2 flight track marked; 532 nm lidar backscatter (b); 9.6 and 94 GHz radar PIA (c); target classification from radar–lidar synergy (d); 9.6 GHz radar reflectivity (e) and mean Doppler velocity (f); and 94 GHz radar reflectivity (g) and mean Doppler velocity (h).





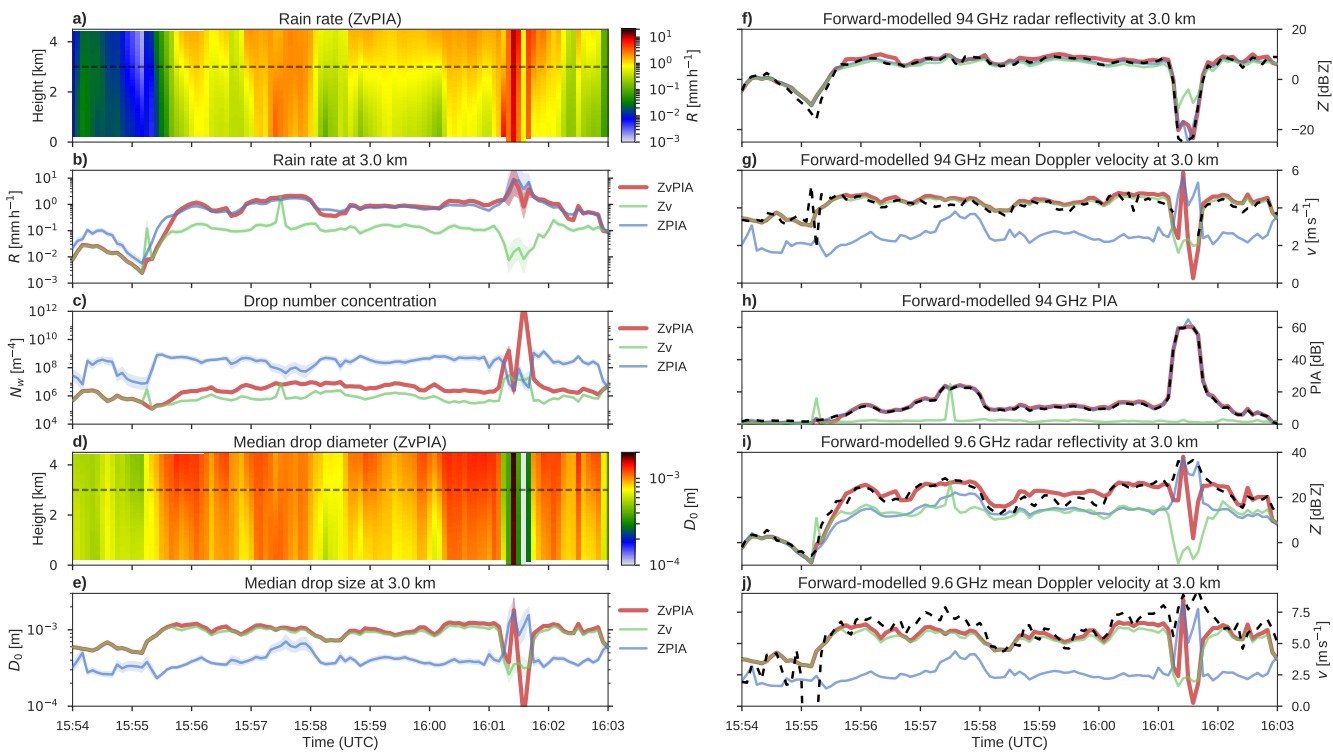

**Figure 4.** Time series of $94\,\mathrm{GHz}$ ZPIA, Zv and ZvPIA retrievals compared for Case 1 between 15:54 and 16:13 on 22 July 2007. Retrieved state and derived variables (left), and forward-modelled radar measurements (right) for the three retrievals are shown at a height of $3\,\mathrm{km}$ above sea level (indicated with a light dashed line in the left-hand scenes), while the full scenes of $R$ (a) and $D_0$ (d) are shown for the ZvPIA retrieval. Shading indicates the $1$-$\sigma$ uncertainty in the retrieved and derived variables. Dark dashed lines (right) indicate the observed radar measurements.



**Figure 5.** Profiles of forward-modelled 94 GHz radar reflectivity (a), mean Doppler velocity (b), PIA (c), retrieved rain rate (d), number concentration parameter (e) and median drop size (f) for ZPIA, Zv and ZvPIA retrievals at 15:58:00 on 22 July 2007. Shading indicates the 1-$\sigma$ uncertainty in the retrieved and derived variables.



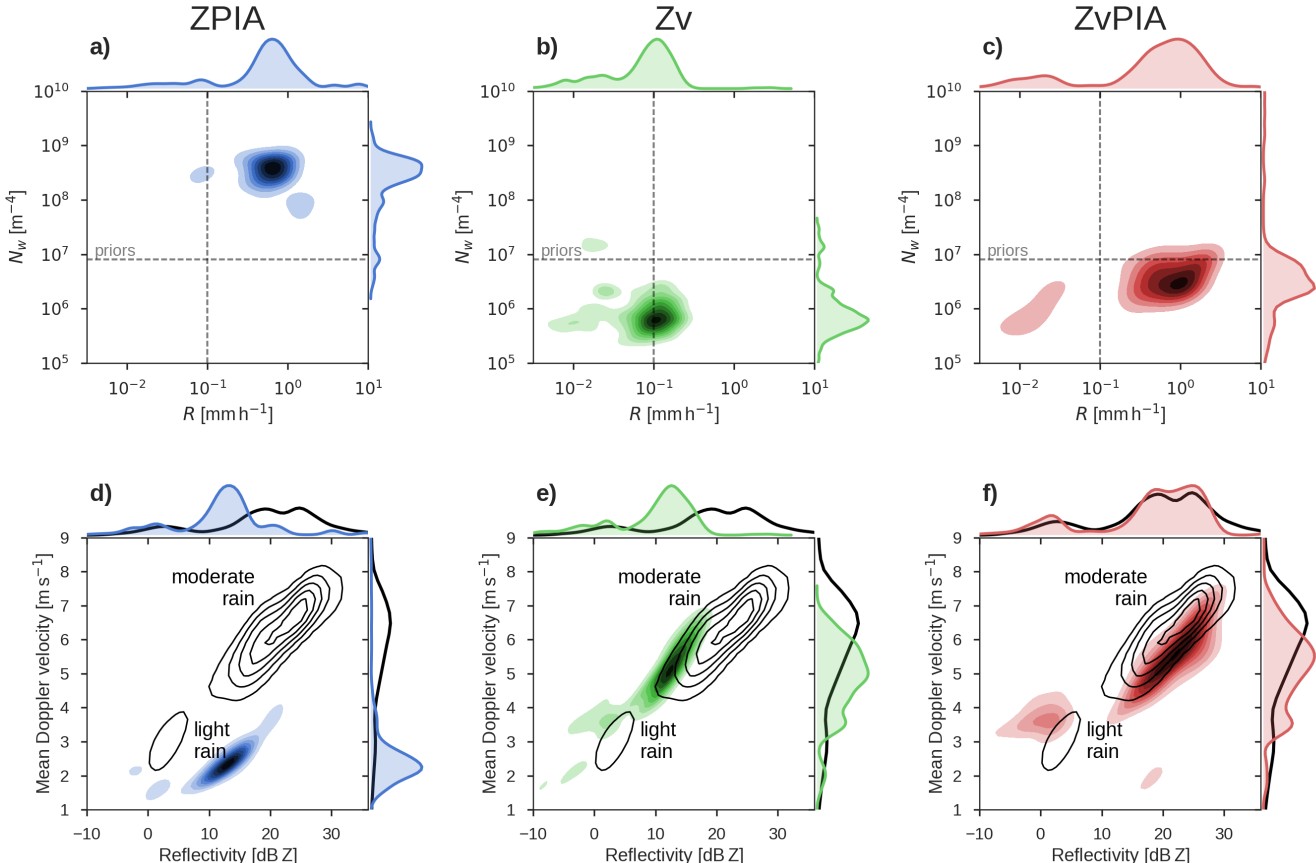

**Figure 6.** Joint (filled contours) and univariate (curves) kernel density estimation histograms of retrieved rain state variables $R$ and $N_w$ (a–c) and forward-modelled EDOP measurements (d–f) for ZPIA (a and d), Zv (b and e) and ZvPIA (c and f) rain retrievals during Case 1 on 22 July 2007. Dashed lines indicate the values of the prior state variables used in the retrieval. Black contours indicate the distribution of independent EDOP measurements; the major rain regimes are labelled.





**Figure 7.** Selected measurements made by ER-2 instruments for Case 2 between 13:12:00 and 13:17:30 UTC on 22 July 2007 as part of TC4. The composite cloud image from MAS/MASTER visible channels (a), with the ER-2 flight track marked; 532 nm lidar backscatter (b); 9.6 and 94 GHz radar PIA (c); target classification from radar–lidar synergy (d); 9.6 GHz radar reflectivity (e) and mean Doppler velocity (f); and 94 GHz radar reflectivity (g) and mean Doppler velocity (h).



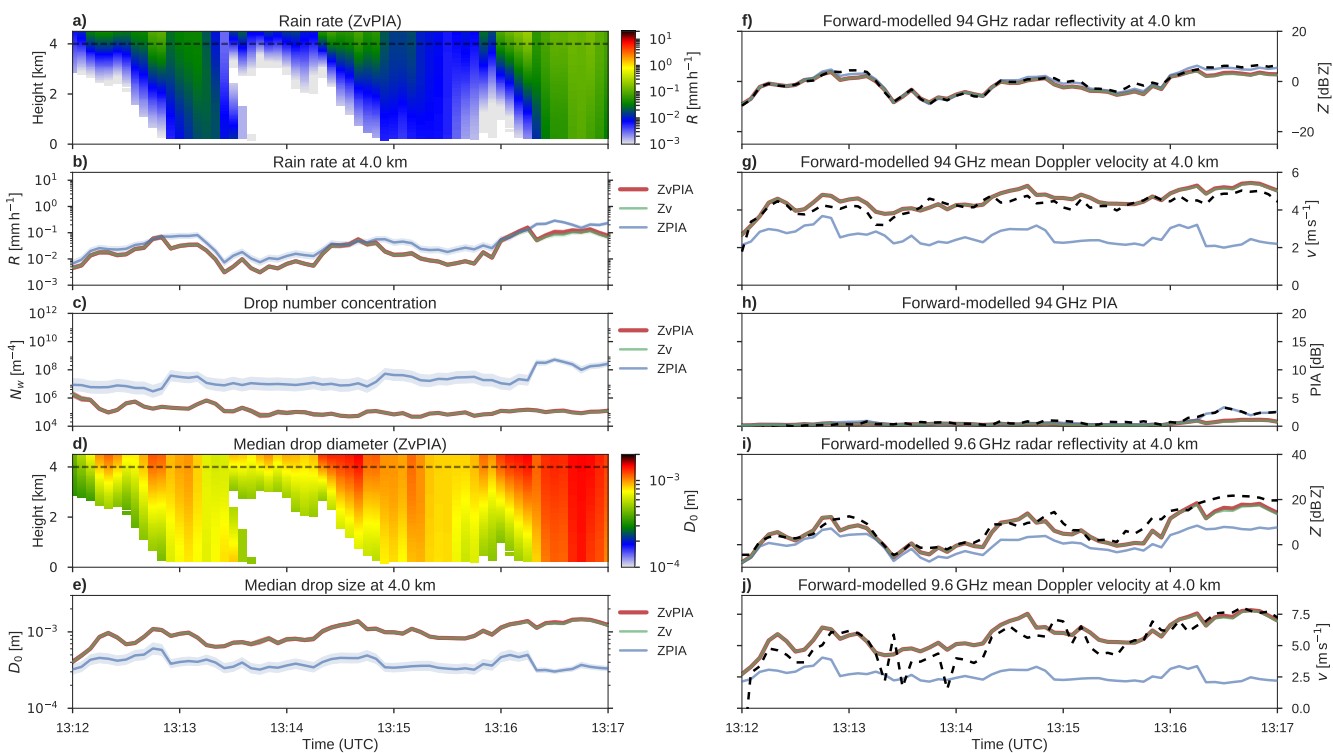

**Figure 8.** Time series of $94\,\mathrm{GHz}$ ZPIA, Zv and ZvPIA retrievals compared for Case 2 between 13:12 and 13:17 on 22 July 2007. Retrieved state and derived variables (left), and forward-modelled radar measurements (right) for the three retrievals are shown at a height of $4\,\mathrm{km}$ above sea level (indicated with a dashed line in the left-hand scenes), while the full scenes of $R$ (a) and $D_0$ (d) are shown for the ZvPIA retrieval. In this case the observed PIA is negligible, so the ZvPIA retrieval has no more information than the Zv retrieval and the two lines are overlaid. Shading indicates the 1-$\sigma$ uncertainty in the retrieved and derived variables. Black dashed lines indicate the observed radar measurements for comparison with the retrievals.

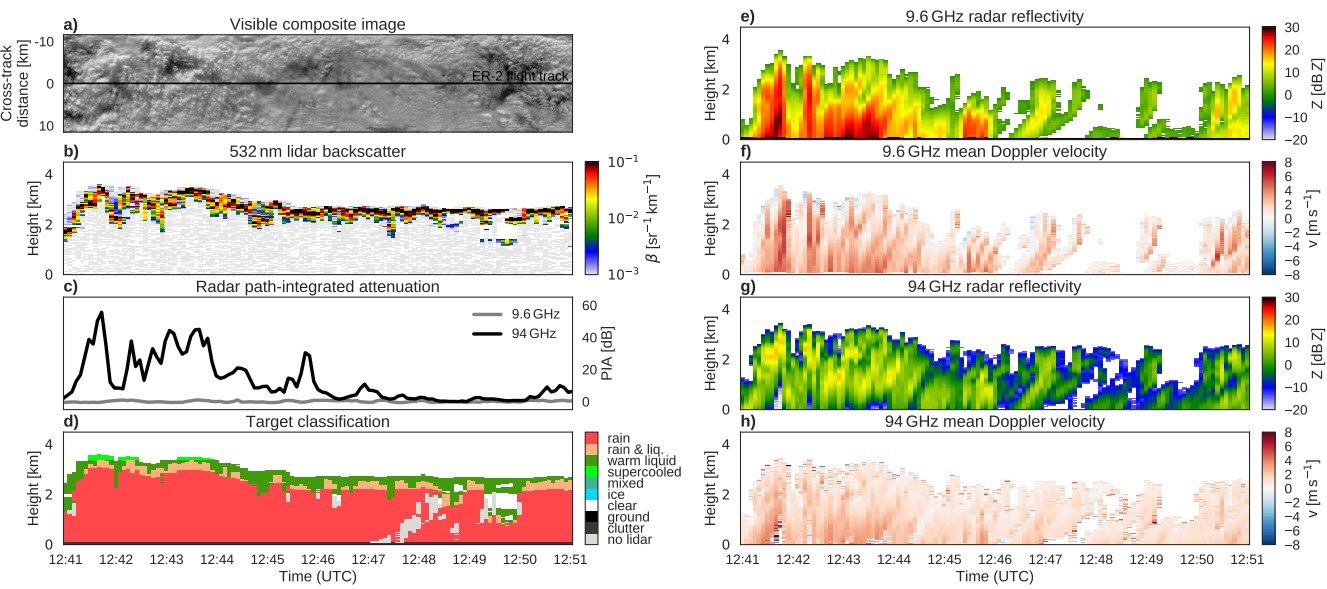

**Figure 9.** Selected measurements made by ER-2 instruments for Case 3 between 12:41 and 12:51 UTC on 29 July 2007 as part of TC4. Composite cloud scene (a) from MAS/MASTER visible channels, with the ER-2 flight track marked; 532 nm lidar backscatter (b); 9.6 and 94 GHz radar PIA (c); target classification from radar–lidar synergy (d); 9.6 GHz radar reflectivity (e) and mean Doppler velocity (f); and 94 GHz radar reflectivity (g) and mean Doppler velocity (h).





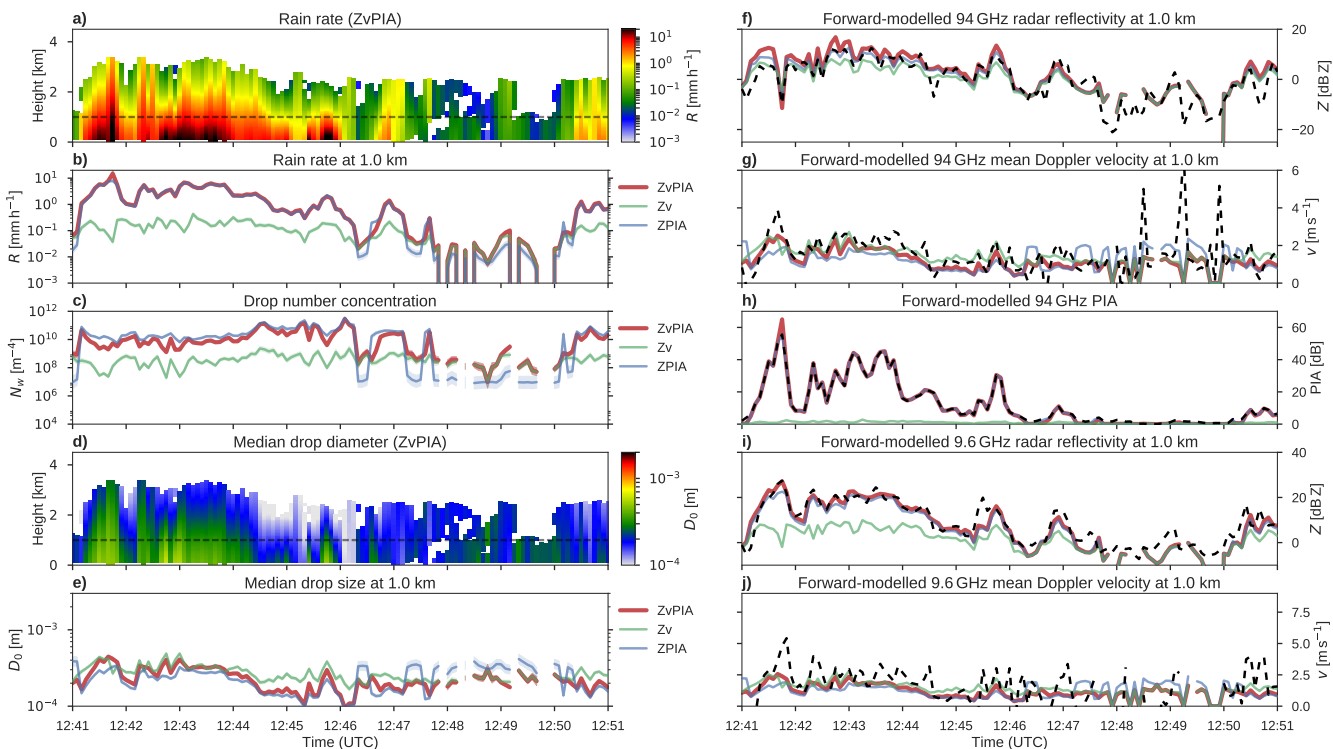

**Figure 10.** Time series of ZPIA, Zv and ZvPIA $94\,\text{GHz}$ retrievals for Case 3 on 29 July 2007. Retrieved state and derived variables (left), and forward-modelled radar measurements (right) for the three retrievals are shown at a height of $1\,\text{km}$ above sea level (indicated with a light dashed line in the left-hand scenes), while full scenes of $R$ (a) and $D_0$ (d) are shown for the ZvPIA retrieval. Shading indicates the $1$-$\sigma$ uncertainty in the retrieved and derived variables. Dark dashed lines (right) indicate the observed radar measurements.





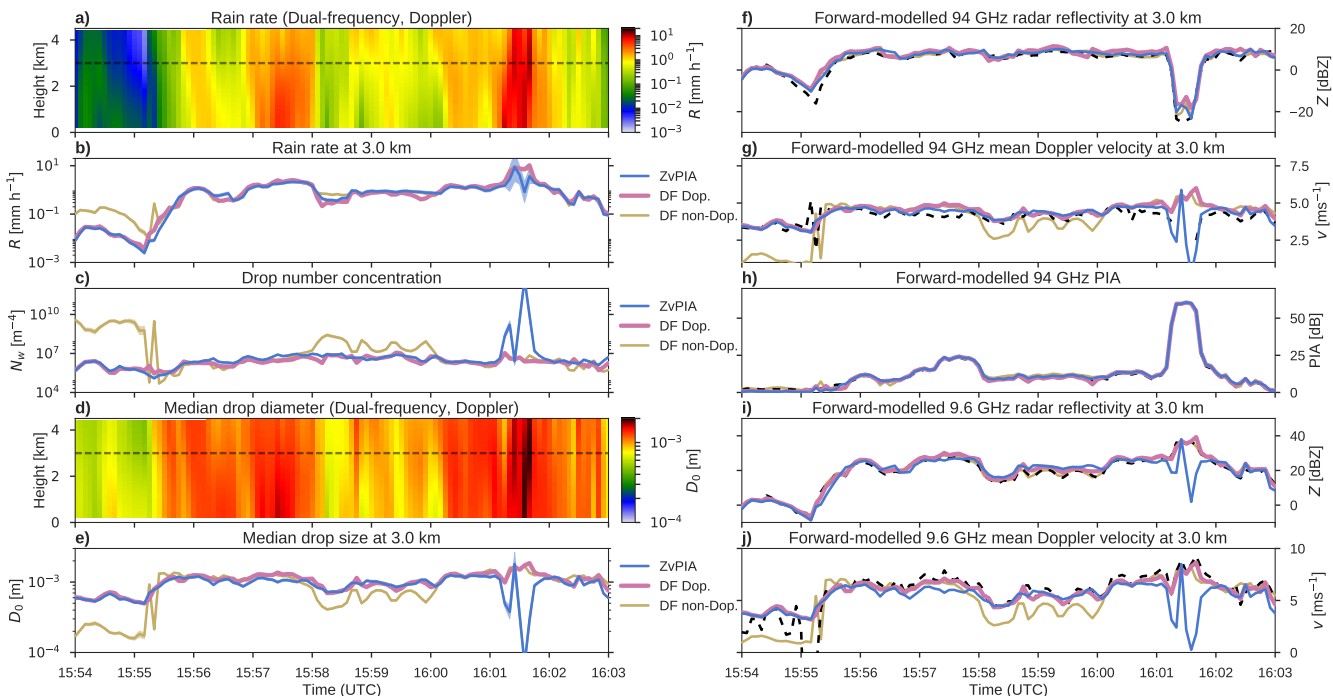

**Figure 11.** Time series of dual-frequency (DF) retrievals with and without Doppler, compared against the $94\,\mathrm{GHz}$ ZvPIA retrieval for Case 1 on 22 July 2007. Retrieved state and derived variables (left), and forward-modelled radar measurements (right) for the three retrievals are shown at a height of $3.0\,\mathrm{km}$ above sea level (indicated with a light dashed line in the left-hand scenes); while the full scenes of $R$ (a) and $D_0$ (d) are shown for the dual-frequency Doppler retrieval. Shading indicates the $1\text{-}\sigma$ uncertainty in the retrieved and derived variables. Dark dashed lines (right) indicate the observed radar measurements.