# Peer review of "Improved rain-rate and drop-size retrievals from airborne Doppler radar"

_Atmospheric Chemistry and Physics, 2017_

## Referee Comment (RC2) · Anonymous Referee #2 · 11 May 2017

This paper uses airborne radar data from the TC4 field experiment to evaluate 94 GHz Doppler radar retrievals of rainfall. Several versions of retreiveal are compared using reflectivity (Z), Doppler (D), and Path Integrated Attenuation (PIA), then only Z-D, and also Z-PIA. Taking Z-D-PIA is truth the authors explore the information available in each of the other two methods. Z-PIA generally performs well but for the lightest rain V-D performs better. The full retrieval Z-D-PIA has some skill in determining drop number concentrations. These observations will be available from EarthCARE and will help with further constrain light rainfall beyond what CloudSat can do.

The paper is well written with no observable flaws in the analysis. My specific comments below are minor. I have some recommendations regarding references. The uncertainties assumed in the optimal estimation need to be justified. I think using an a-priori in this retrieval is unjustified but others would disagree with me so I won't fight

the point.

I recommend returning to the authors for minor revisions.

Line 1: remove second 'of'

Page 1, line 18: also A. Behrangi, 2012

Page 8, line 19: Lebsock et al., 2013 is a better reference than Lebsock et al., 2011

Line 20: Stephens et al. only references intensity. Cite Abel and Boutle, 2012 for the DSD.

Page 2: line 5. I would mention the GPM DPR before bringing up cloudsat. It has increased sensitivity (12 dbz) vs PR (17 dbz).

Page 2: Lines 14-16. CloudSat actually does not use the SRT (as in the Meneghini 1983 definition) method in its operation products. Instead a look-up-table of normalize surface cross section as a function of wind speed and SST from ECMWF analysis. But Lebsock et al. (2011) do use the SRT for cloud/precipitation water path estimates from CloudSat. The SRT can provide a superior estimate when the length scale of precipitation is short. In the next release of cloudsat precipitation products (release 05) cloudsat will use a hybrid method combining the LUT and SRT techniques each used where appropriate.

Page 2, line 30: although these approaches have been challenged in practice due to the spectral dependence of the non-uniform-beam-filling and multiple scattering affecting the two frequencies.

Page 3: Lines 27/28: should be Lebsock et al., 2011

Equation 1: What is the structure of $J_c$? What is the structure of B? This is where all the magic happens. Also it's really hard to justify a prior in this retrieval. Why do you feel that you need one? And how do you justify it? The prior variance had better be large. Rodgers was doing sounding where you might actually have a moderately

reasonable prior constraint. This approach shouldn't A prior of 0.1 mm/hr is going to make it hard to retrieve 10 mm/hr.

Page 6, Line 28. I think the assumption is a necessary one but I think that the justification has less to do with rain rate and more to do rain type. For broad areas of stratiform precipitation I would suppose the invariance is most appropriate but that for any type of convection (light or heavy rain) I'm not so sure.

Page 7, Line 10: This value of k is not consistent with your retrieved rain rate DSD in that bin though is it? I'm not that concerned about this but if I'm correct you should make a note of it.

Page 8, Line 5: add citation for the models that use for attenuation. There are many.

Page 8, Line 15: I'm really skeptical of this Matrosov approach. It really depends strongly on steady state rainfall. How often does this happen? In practice in CloudSat data I see that the stuff that really fully attenuates the radar is the convection, where I just wouldn't trust this assumption.

Table 2: a little discussion about where these numbers come from is required. 0.3 dB would be a very good estimate of PIA. From the SRT method the instrument noise alone is probably close to this value. Is the 3 dBZ including the uncertainty in your DSD – Z relationship? Another thing to consider is whether this 3 dBZ uncertainty should be constant with height. In reality uncertainty should grow with depth into the column (e.g. Lebsock and L'Ecuyer, 2011) because any errors you make in your forward modeled attenuation in the range volumes up high is compounded as you descend into the rain column. This becomes an issue when the PIA is order 20 dB like the cases you explore later in the paper.

Fig 5a and elsewhere: I can't understand why the Zv retrieval is not matching the radar reflectivity. Doesn't Zv mean that both radar reflectivity and mean Doppler are used to constrain the solution. It looks like only Doppler is used in these plots.

Conclusions: The idea that mean Doppler will help with light rainfall retrievals over land is good but I worry that you often won't know whether you are looking at light rainfall or heavier rainfall that appears light because it is attenuated. This process will need to be automated in a retrieval algorithm and it doesn't' seem straightforward. Also, the signal in the Doppler is not all that large relative to the uncertainties that are expected.

Conclusions: Won't Doppler be affected by Multiple scattering for EarthCARE? I haven't done any calculations of this but I wonder how this might complicate things.

Refs

A. Behrangi, M. Lebsock, S. Wong, B. Lambrigtsen, 2012: On the quantification of oceanic rainfall using space-born sensors, J. Geophys. Res., 117(D20), D20105.

Lebsock, M., H. Morrison, and A. Gettelman (2013), Microphysical implications of cloud-precipitation covariance derived from satellite remote sensing, J. Geophys. Res. Atmos., 118, doi:10.1002/jgrd.50347.

Abel, S. J. and Boutle, I. A. (2012), An improved representation of the raindrop size distribution for single-moment microphysics schemes. Q.J.R. Meteorol. Soc., 138: 2151–2162. doi:10.1002/qj.1949

---

## Referee Comment (RC3) · A. Protat (Referee) · 22 May 2017

Review of "Improved rain-rate and drop-size retrievals from airborne and spaceborne Doppler radar", by S. Mason et al.

This paper presents a new variational retrieval technique to estimate rain rate and information about the drop size distribution from airborne 94 GHz Doppler cloud radar data. This work is important given the perspective of the launch of the first 94 GHz Doppler cloud radar in space. I recommend this paper be accepted provided that the comments below are addressed prior to publication. My only "major" comment

General comments :

1. Page 11, line 5: Since you don't have independent measurements to validate your

algorithm, you have elected to use Z9.6 and BV9.6 as an indirect validation, which is OK but is certainly a limitation in the paper. It would be great to make the most out of it. So I wonder why you have decided not to show the whole vertical distribution (and maybe difference plots). That would characterize the errors in a more exhaustive way. A major improvement to the paper would be to demonstrate that with the limited validation you have, you can show that the vertical distribution is captured.

2. Discussion on case study 2, Fig. 8: No comment on the vertical distribution of D0 for that difficult case? In evaporation conditions is it realistic to see D0 decreasing at lower altitudes? The fact that Nw is held constant should be an issue here, as one might expect Nw to strongly diminish lower in height, and maybe Do increase due to removal of the smaller drops evaporating? That is one illustration of my general comment 1. You need to show the vertical distribution, not a selected height, because it does not tell the full story.

3. Page 17, lines 23-24: You say that in this study you have used measurements to investigate the prospects for improved global rain rate retrievals from spaceborne Doppler radar. However in order to study that more completely you could have chosen to simulate from these aircraft observations what EarthCARE would actually measure, and whether you would be able to get similar results between degraded spaceborne measurements and higher-resolution higher-quality airborne measurements. I feel this piece of work is missing to really make that claim. In that respect I would remove mention of "spaceborne" in the title of the paper, not to give the impression that you are actually improving satellite rain rates with this technique, as I don't think you showed that.

Specific comments :

1. Page 1, line 11: separate "between" and "light".

2. Page 3, line 10, " . . . unattenuated walvelength . . .". This is incorrect, there is still large attenuation at X-band. This needs to be rephrased and errors associated with

potential X-band attenuation assessed.

3. Page 3, line 15, "dual-radar" : do you mean dual-frequency radar ?

4. Section 2.3.4 title : What about 94 GHz attenuation due to graupel or hail ? Something needs to be said about that in this paper.

5. Page 8, line 5 : "assume multiple scattering effects are negligible . . .". You should probably explain why you think that is reasonable (very small beamwidth). A comment is also needed to explain that this would need to be done in a spaceborne application.

6. Page 8, line 24: "as 3 dBZ, and as 0.5ms$-1$ for mean Doppler. . .". This seems large for reflectivity and too small for Doppler velocity. Vertical air motions can be 1-2 ms-1 in the lower troposphere easily in the clouds you are interested in ... That brings up a question you need to address (sorry . . .): how sensitive to this value are the results ?

7. Page 8, line 27, " .. by liquid water . . .". What about melting ice, graupel, hail ?

8. Page 14, line 31-32: What about at lower height ? I would expect that if the Nw assumption is not satisfied lower down in the evaporating area, then D0 should have more errors and then Z(9.6 GHz) would be less good. It would actually be more interesting to show the whole vertical distribution instead of extracting one height for all these plots to demonstrate if the Nw assumption does create discrepancies on the vertical distribution of Z9.6. Hence my general comments 1 and 2.

9. Section 4.3.1, title: I think you could go to 12:46 for your comparisons? Why did you stop at 12:45 ?

Good luck with the review.

Alain Protat, Melbourne, May 2017.

---

## Author Comment (AC1) · 6 Jul 2017

We thank the reviewers for their constructive comments, and hope that our responses have helped to improve the paper.

A common thread across the reviews was a request for more justification of the retrieval of a height-invariant $N_w$, and for evaluation of the retrievals through the vertical profile. In response we have added Figs. 5, 9 & 12 evaluating the averaged vertical profile of retrieved and forward-modelled variables in key precipitation regimes: moderate stratiform rain (case 1), light stratiform rain with strong evaporation (case 2), and moderate warm rain (case 3).

In evaluating the instances where the constant-$N_w$ representation was not able to reproduce the profile of 9.6 GHz radar observations, we thought it worthwhile to add a demonstration of the retrieval in which $N_w$ is represented as a linear gradient (Section 6). We show that introducing another degree of freedom allows us to resolve some of the variations in the DSD through the profile as expected for collision-coalescence, and that these changes lead to an improved ability to forward-model the independent 9.6 GHz radar measurements. This is possible with the high vertical resolution of the airborne radar observations, and therefore worth demonstrating, but we do not necessarily anticipate retrieving a vertical profile of $N_w$ from EarthCARE, which will have coarser 500m vertical resolution.

**Major comments**

**1. I wonder how the absorption by liquid water clouds is handled. Liquid phase clouds below the melting layer will contribute to the total PIA. Their contribution can be substantial especially for lighter rainfall. Neglecting cloud absorption will result in overestimation of PIA due to rain. Cloud base heights can be significantly lower than the melting layer.**

Radar attenuation by liquid cloud water is estimated within the forward model as for liquid rain water; however the detection of liquid clouds below the melting layer is difficult from above, where lidar tends to be extinguished and radar is dominated by the larger drops. We can be confident that the rain will dominate the radar attenuation, but it's true that this will be an upper estimate of the attenuation attributed to rain, and could include some fraction due to unseen clouds.

Changes:

This uncertainty has been made more explicit in Section 2.2 Target Classification; the relevant paragraph now includes an additional statement:

> As a result of the uncertain presence of liquid clouds within rainy profiles, the path-integrated attenuation of the radar that is attributed to rain may be partially due to undiagnosed liquid cloud.

**2. Assuming that Nw is constant with height does not account for drop collision-coalescence, evaporation and breakup processes. It is a rather heavy assumption and it needs more justification.**

Retrieving constant Nw for each profile is an improvement over assuming Nw is constant everywhere; however, we agree that the representation of Nw is not expected to be borne out physically in many cases. Therefore the constant Nw may be best interpreted as a profile-averaged Nw; but this warrants further discussion in the text.

There may indeed be sufficient information in some cases to retrieve more complex representations of the profile of Nw, may better fit our understanding of microphysical processes, and we agree this was important to discuss in more detail.

Changes:

To better justify the decision to retrieve constant Nw, we have added the following discussion in Section 2.3.3 defining the rain state variables:

> Additional state variables increase the degrees of freedom of the retrieval, and require more information from observational variables to constrain the

> retrieval. Therefore we retrieve a single value of Nw for each profile, with the physical interpretation of representing Nw as constant with height, or as the vertically-averaged value. The representation of Nw as constant with height is not expected to be borne out in cases where evaporation or collision-coalescence processes modify the drop number concentration through the vertical profile.

To explore the possibility of retrieving more detailed profiles of Nw, we have added Section 6, in which we retrieve Nw as a linear gradient for the warm rain case. The vertical profile of moderate warm rain is evaluated against the 9.6-GHz radar variables, and we show that by allowing the additional degree of freedom for a linear gradient of Nw we can better represent the mean Doppler velocity toward the surface, improving the fit to independent radar measurements, as well as (qualitatively) resolving the drop growth, and decreasing drop concentration, toward the surface we would expect from collision-coalescence processes in this context.

In the discussion and conclusions, we have added a paragraph to discuss the representation of Nw, summarising the findings of Section 6.

**Other comments**

**1. Do you account for changes in raindrop terminal velocities with altitude as air density changes?**

Terminal velocity is corrected for air density through the profile, however this had been unclear in the methods section ("... corrected for air density").

Changes:

The description of mean Doppler velocity now says "…scaled to account for air density changes with altitude."

**2. From the text I understood that gaseous attenuation is calculated from model profiles of temperature and humidity. In stratiform rain, however, relative humidity is often 90-95% and if model profiles suggest lower humidity (e.g., the model does not forecast rain in a particular pixel) the water vapor absorption contribution in PIA can be underestimated.**

Yes, the relative humidity profile from the model is not updated in the presence of rain. In practice for the cases studied here, we have confirmed that the relative humidity in the model data already exceeds 90% for most of the rainy part of the vertical profile.

However the situation seems likely to occur at times, so the water vapour contribution to gaseous attenuation could be set in the algorithm to be the larger of the re-analysis and 90%.

**3. The statement that the gradient method requires an assumption of constant rain rate with height is misleading. In fact it requires an assumption that non-attenuated reflectivity changes are small compared to changes due to attenuation. This method provides an average rain rate in the height interval which is used to calculate the gradient.**

Thank you for this clarification.

Changes:

The text now reads:

> Both approaches are implemented simultaneously, so that whereas the gradient method of Matrosov et al. (2007) is applied only at moderate to heavy rain rates wherein it can be assumed that the gradient of apparent radar reflectivity is dominated by attenuation, within the CAPTIVATE variational scheme the gradient of R and k can be estimated simultaneously from the profile of radar reflectivity and PIA.

**4. When using 9.6 GHz data, do you account for rain attenuation at this frequency? Estimates show that attenuation at X-band at 10 mm/h at nadir pointing and in a 4 km thick layer could be around 1.3 dB. In addition to that the melting layer attenuation will add a contribution, which cannot be neglected.**

The attenuation of the 9.6 GHz radar due to both rain and the melting layer are included in the forward model.

Changes:

In Section 2.3.4 on the melting layer, the corresponding value of k for X-band radar from Matrosov (2008) are now given.

In Section 2.3.5 on the radar forward model, we now mention both frequencies explicitly.

**5. How well are radar beams at X and W bands matched? The DDV measurements are very sensitive to beam mismatches.**

The X- and W-band radars aboard ER-2 have been previously used for DDV measurements (e.g. Tian et al. 2007) from CRYSTAL-FACE, and by averaging to 5-second (1 km) along-track we expect that the matching between the two radar sampling volumes is strongly correlated despite differences in beam widths.

In terms of pointing error, visually there is no appearance of significant features being poorly correlated, and we do not correct for any known differences in radar pointing.

**6. I believe the reference to Matrosov et al. (2008) in JAS in line 10 on page 7 for eq. (6) is wrong, it should be the reference to Matrosov (2008) IEEE TGARS, 1039-1047, doi: 10.1109/TGRS.2008.915757 This equation provides two-way attenuation. Your assumption of Xm=1 km actually corresponds to the melting layer thickness of 0.5 km, which accidentally is about right as melting layers often have thicknesses of around 0.5 km. Please correct the reference and the Xm definition.**

Thank you for catching this: this was indeed the paper we intended.

Changes:

We have corrected the reference.

The definition of Xm is updated.

**7. Figure 3. It appears that PIA is saturated at values lower than 60 dB, but the text says it is 65 dB.**

Fixed.

**8. Did you estimate what is the uncertainty of using the Mie theory instead of calculations for oblate raindrops?**

The T-matrix method is also implemented in CAPTIVATE, so we calculated the effect of including oblate drops at larger diameters. Using Thurai et al. (2007) and Zhang et al. (2001) for the axial ratios of raindrops, the error in total backscatter due to assuming spherical drops is around 5% for a DSD with D0 of 1.5mm, which is roughly the largest D0 retrieved in this study; and much less for smaller median drop diameters. Errors in extinction are less than 2%.

Changes:

These uncertainties are now noted in Section 2.3.5 when radar reflectivity and extinction are described.

**9. Figure 4 shows PIA-based retrievals also for the period when the W-band signal was completely extinguished (between 16:02 and 16:03 UTC), so PIA was not available. How it is possible?**

When the radar is fully attenuated and the surface backscatter signal is indistinguishable from the noise, the PIA signal saturates, but is still available. This effect is accounted for in the forward-model, so that while the saturated PIA no longer provides an accurate estimate of the total attenuation in the profile, the information that the radar is

extinguished still provides some information for the retrieval.

**Editorial comments**

**1. Since you use natural logarithms in (4), (5), (9) and Table 1, you should change "log" to "ln".**

Fixed

**2. Page 8 line 24 and Table 2: 3 dBZ -> 3 dB (relative units).**

Fixed.

**3. Table 2. You do not measure Z as it is given in (7), but rather you measure attenuated Z.**

True. The apparent reflectivity Za is now introduced in this section, and used in Table 2.

---

## Author Comment (AC2) · 6 Jul 2017

We thank the reviewers for their constructive comments, and hope that our responses have helped to improve the paper.

A common thread across the reviews was a request for more justification of the retrieval of a height-invariant Nw, and for evaluation of the retrievals through the vertical profile. In response we have added Figs. 5, 9 & 12 evaluating the averaged vertical profile of retrieved and forward-modelled variables in key precipitation regimes: moderate stratiform rain (case 1), light stratiform rain with strong evaporation (case 2), and moderate warm rain (case 3).

In evaluating the instances where the constant-Nw representation was not able to reproduce the profile of 9.6 GHz radar observations, we thought it worthwhile to add a demonstration of the retrieval in which Nw is represented as a linear gradient (Section 6). We show that introducing another degree of freedom allows us to resolve some of the variations in the DSD through the profile as expected for collision-coalescence, and that these changes lead to an improved ability to forward-model the independent 9.6 GHz radar measurements. This is possible with the high vertical resolution of the airborne radar observations, and therefore worth demonstrating, but we do not necessarily anticipate retrieving a vertical profile of Nw from EarthCARE, which will have coarser 500m vertical resolution.

**Line 1: remove second 'of'**

Done.

**Page 1, line 18: also A. Behrangi, 2012**

Added.

**Page 1, line 19: Lebsock et al., 2013 is a better reference than Lebsock et al., 2011**

Added.

**Page 1, Line 20: Stephens et al. only references intensity. Cite Abel and Boutle, 2012 for the DSD.**

Added.

**Page 2: line 5. I would mention the GPM DPR before bringing up cloudsat. It has increased sensitivity (12 dbz) vs PR (17 dbz).**

This brief timeline of spaceborne cloud and precipitation radars was intended to introduce the capabilities of CloudSat's 94-GHz radar for sensing the vertical profile of light rain, including evaporation and drizzle, in contrast to precipitation radars; for brevity, and because CloudSat followed TRMM historically, we prefer not to introduce GPM here.

**Page 2: Lines 14-16. CloudSat actually does not use the SRT (as in the Meneghini 1983 definition) method in its operation products. Instead a look-up-table of normalized surface cross section as a function of wind speed and SST from ECMWF analysis. But Lebsock et al. (2011) do use the SRT for cloud/precipitation water path estimates from CloudSat. The SRT can provide a superior estimate when the length scale of precipitation is short. In the next release of cloudsat precipitation products (release 05) cloudsat will use a hybrid method combining the LUT and SRT techniques each used where appropriate.**

Thanks for making this distinction; we have clarified that that the surface cross-section can be estimated both from the SRT and from the surface wind speed and temperature.

The statement now reads:

> The path-integrated attenuation (PIA) can be estimated from the the ocean surface backscatter relative to either nearby clear-sky profiles (Meneghini 1983), or calculated from sea surface wind speed and temperature (Haynes 2009). Estimates of PIA are used in the rain retrieval algorithms of both TRMM (Iguchi 2000, Meneghini 2000) and CloudSat (L'Ecuyer 2002, Haynes 2009, Lebsock 2011) over the ocean...

**Page 2, line 30: although these approaches have been challenged in practice due to the spectral dependence of the non-uniform-beam-filling and multiple scattering affecting the two frequencies.**

That is interesting; however, notwithstanding these challenges, GPM rain retrievals are intended to use dual-frequency radar to retrieve two parameters of the DSD, and a more detailed discussion here is not required to justify the Doppler approach. With some minor changes, this sentence now reads:

> The recent global precipitation measurement mission (GPM; Hou et al. 2014), with the first dual-frequency radar in space, in intended to exploit differences in non-Rayleigh scattering at 35-GHz and 14-GHz to better constrain the rain DSD over land and ocean (Rose et al. 2006).

**Page 3: Lines 27/28: should be Lebsock et al., 2011**

Updated.

**Equation 1: What is the structure of Jc? What is the structure of B? This is where all the magic happens. Also it's really hard to justify a prior in this retrieval. Why do you feel that you need one? And how do you justify it? The prior variance had better be large. Rodgers was doing sounding where you might actually have a moderately reasonable prior constraint. This approach shouldn't A prior of 0.1 mm/hr is going to make it hard to retrieve 10 mm/hr.**

Thank you. We have added more detail and references to describe the terms in Equation 1.

As for a prior R, we do apply it with a large prior variance, so that its influence is small

for a well-constrained retrieval. However, as we are demonstrating under-constrained ZPIA and Zv retrievals, the choice of prior does have some influence on the results; this sensitivity to the prior and the double-minimum problem is demonstrated in Section 3.

Changes:

To describe Jc and B in more detail, we now write:

> . . .and B is the error covariance matrix of xa in which the diagonal elements are the error variances of x; and Jc(x) provides the capability to apply flatness and smoothness constraints to reduce the effect of observational noise on the state vector (Twomey 1977). Additionally, profiles of retrieved variables can be represented smoothly as a set of cubic spline basis functions (Hogan 2007, and in Section 2.3.3).

To better explain the use of a prior R, we have added the following:

> For all retrievals a prior R of 0.1 mm/h is used. While a prior R is not strictly necessary, it is applied in combination with a large prior uncertainty such that the retrieved R should be relatively insensitive to the prior unless the retrieval is poorly constrained by observations. We note that this value for the prior variance implies that before the measurements are taken we assume there is a 44

**Page 6, Line 28. I think the assumption is a necessary one but I think that the justification has less to do with rain rate and more to do rain type. For broad areas of stratiform precipitation I would suppose the invariance is most appropriate but that for any type of convection (light or heavy rain) I'm not so sure.**

This distinction is important, and we had over-stated this assumption here; in practice, R is always represented as the basis functions of a cubic spline to allow for vertical variability.

This statement now reads:

> While in moderate stratiform rain R is often close to invariant with height (e.g. Matrosov 2007), processes such as evaporation in the lower atmosphere, and collision–coalescence in warm clouds, will lead to significant variation with height in many contexts. R is therefore represented as the coefficients of a cubic spline basis function with n elements (Hogan 2007); this has the effect of ensuring the vertical profile of R is smoothly varying and continuous with height, and also reducing the number of terms in the state vector.

**Page 7, Line 10: This value of k is not consistent with your retrieved rain rate DSD in that bin though is it? I'm not that concerned about this but if I'm correct you should make a note of it.**

Matrosov (2008) assumed a Marshall–Palmer distribution in the rain below the melting layer, so there is indeed a mismatch in the DSD assumptions in the rain and in the melting layer,

This is now explicit in the text:

The estimated attenuation through the melting layer is based on a Marshall–Palmer DSD for the rain below the melting layer (Matrosov 2008), and is not modified to match the retrieved DSD in the profile.

**Page 8, Line 5: add citation for the models that use for attenuation. There are many.**

Citation to Liebe (1985) added.

**Page 8, Line 15: I'm really skeptical of this Matrosov approach. It really depends strongly on steady state rainfall. How often does this happen? In practice in CloudSat data I see that the stuff that really fully attenuates the radar is the convection, where I just wouldn't trust this assumption.**

In response to comments from reviewer 1 we have clarified that the necessary assumption for this approach is not that the rain rate is constant, but only that attenuation dominates any observed gradient in apparent radar reflectivity. Subsequently the estimated R from this method is related to the average over the gates in question.

**Table 2: a little discussion about where these numbers come from is required. 0.3 dB would be a very good estimate of PIA. From the SRT method the instrument noise alone is probably close to this value. Is the 3 dBZ including the uncertainty in your DSD– Z relationship? Another thing to consider is whether this 3 dBZ uncertainty should be constant with height. In reality uncertainty should grow with depth into the column (e.g. Lebsock and L'Ecuyer, 2011) because any errors you make in your forward modeled attenuation in the range volumes up high is compounded as you descend into the rain column. This becomes an issue when the PIA is order 20 dB like the cases you explore later in the paper.**

In practice the results of the ZvPIA retrieval are fairly insensitive to the uncertainties, with the exception of the under-constrained Zv and ZPIA retrievals. The uncertainty in these variables should include both measurement and forward-model error, so in response to several reviewer comments we now use more physically reasonable values: 3 dB for radar reflectivity,

1.0 m/s for mean Doppler velocity,

0.5 dB for PIA

Changes:

All figures have been re-run with the updated uncertainties. The changes are small, especially for the ZvPIA retrievals that are well-constrained by observations. Table 2 is updated, and the values of the uncertainties updated in the text.

The following discussion has been added:

> We have found that the weighting of errors between radar reflectivity and PIA is quite important for the retrieved rain rate, and that if only instrument errors are included the retrieval is not sufficiently constrained by PIA. This is believed to be because attenuation affects all forward-modelled radar

reflectivity measurements in the same way, leading to them having strong error correlations. Error correlations are not accounted for in the R matrix, since they are profile-dependent and difficult to estimate, which can lead to the radar reflectivity measurements being over-weighted in the retrieval. To overcome this, we take the common approach (e.g. Weston et al. 2014) of inflating the reflectivity errors (and in our case somewhat reducing the errors in PIA) to better balance the information coming from the reflectivity profile and from PIA.

**Fig 5a and elsewhere: I can't understand why the Zv retrieval is not matching the radar reflectivity. Doesn't Zv mean that both radar reflectivity and mean Doppler are used to constrain the solution. It looks like only Doppler is used in these plots.**

The Zv retrieval does make use of both radar reflectivity and mean Doppler velocity, but is under-constrained, and therefore prone to bimodal errors as illustrated in Section 3.

In this case the Zv retrieval starts from the prior R of 0.1 mm/h, and stays in a low-R state. While D0 is constrained by the Doppler, without PIA R is under-estimated, and therefore—without including attenuation in the forward-model—the retrieval has difficulty resolving the gradient of Z. Conversely, starting from a higher prior R improves the performance of the Zv retrieval in this case, but would lead to high-R retrievals in the light rain profiles.

This sensitivity to the prior is not evident for the ZvPIA retrievals, which are well constrained.

[Figure]

**onclusions: The idea that mean Doppler will help with light rainfall retrievals over land is good but I worry that you often won't know whether you are looking at light rainfall or heavier rainfall that appears light because it is attenuated. This process will need to be automated in a retrieval algorithm and it doesn't seem straightforward. Also, the signal in the Doppler is not all that large relative to the uncertainties that are expected.**

It's true that this will be a challenge, but we think the results shown here are promising. The synthetic profile in Section 3, and the moderate rain profiles with PIA   20 dB in Case 1 showed that the Zv retrieval can sometimes be used to distinguish weakly and strongly attenuated profiles based on the profiles of Z and v; however, it's true that this will not always be sufficient in practice.

It may be necessary to estimate PIA over land (even with large measurement error) to help identify strong attenuation, or to assume Nw in the retrieval, so that t. A simple test for weakly attenuated profiles might be to identify profiles in which the maximum reflectivity is less than some threshold.

Changes:

A "further work" statement is added in the third-to-last paragraph on the contribution of Doppler to the retrieval, while the more optimistic statement about retrievals over land in the final paragraph has been removed.

**Conclusions: Won't Doppler be affected by Multiple scattering for EarthCARE? I haven't done any calculations of this but I wonder how this might complicate things.**

Multiple scattering is indeed expected to affect the Doppler for EarthCARE (Battaglia and Tanelli, 2011). We have added the following:

Changes:

The brief mention of multiple scattering in Section 2.4 now reads:

> Multiple scattering effects on radar and lidar backscatter can be estimated within CAPTIVATE using Hogan (2008). Radar reflectivity enhancement due to multiple scattering is especially relevant to spaceborne radar measurements at millimeter wavelengths (Battaglia 2005), and the effects on Doppler radar measurements are expected to include both enhanced spectral broadening and modified mean Doppler velocity (Battaglia 2011); however, with the narrower beam of the airborne radar used in this study we can assume multiple scattering effects are negligible (Battaglia 2007).

In the conclusion we now reiterate the challenges of applying these methods to Earth-CARE, including multiple scattering and measurement error.
* * *

---

## Author Comment (AC3) · 6 Jul 2017

We thank the reviewers for their constructive comments, and hope that our responses have helped to improve the paper.

A common thread across the reviews was a request for more justification of the retrieval of a height-invariant Nw, and for evaluation of the retrievals through the vertical profile. In response we have added Figs. 5, 9 & 12 evaluating the averaged vertical profile of retrieved and forward-modelled variables in key precipitation regimes: moderate stratiform rain (case 1), light stratiform rain with strong evaporation (case 2), and moderate warm rain (case 3).

In evaluating the instances where the constant-Nw representation was not able to reproduce the profile of 9.6 GHz radar observations, we thought it worthwhile to add a demonstration of the retrieval in which Nw is represented as a linear gradient (Section 6). We show that introducing another degree of freedom allows us to resolve some of the variations in the DSD through the profile as expected for collision-coalescence, and that these changes lead to an improved ability to forward-model the independent 9.6 GHz radar measurements. This is possible with the high vertical resolution of the airborne radar observations, and therefore worth demonstrating, but we do not necessarily anticipate retrieving a vertical profile of Nw from EarthCARE, which will have coarser 500m vertical resolution.

**General comments:**

**1. Page 11, line 5: Since you don't have independent measurements to validate your algorithm, you have elected to use Z9.6 and V9.6 as an indirect validation, which is OK but is certainly a limitation in the paper. It would be great to make the most out of it. So I wonder why you have decided not to show the whole vertical distribution (and maybe difference plots). That would characterize the errors in a more exhaustive way. A major improvement to the paper would be to demonstrate that with the limited validation you have, you can show that the vertical distribution is captured.**

Our instinct was to limit the propagation of figures by comparing multiple retrievals on a single axes: hence the comparison of ZPIA, Zv and ZvPIA retrievals at a selected level above sea level; however, we agree that this didn't allow for sufficient evaluation of the vertical distribution.

Plots of the full vertical curtain plots for each retrieval (ZPIA, Zv, ZvPIA) tend to look quite similar, while the difference plots highlight errors at the tops and the bottoms of

the profile and in both cases, we felt these would significantly increase the number of figures and be relatively difficult to interpret.

We have therefore included averaged vertical profiles of observed and forward-modelled radar measurements and retrieved R, D0 and Nw over selected rain regimes for each case. The new Fig. 5 shows the moderate rain profiles of Case 1, Fig. 9 is for all of Case 2, and Fig. 12 shows the moderate warm rain profiles from Case 3. The new Fig. 5 replaces a previous figure showing a selected vertical profile, but illustrates much the same major points.

As discussed in responses to general comment 2 and specific comment 8, this evaluation in the vertical profile allows for some additional insights into how the DSD is resolved with height, and the limits of the constant-Nw assumption.

**2. Discussion on case study 2, Fig. 8: No comment on the vertical distribution of D0 for that difficult case? In evaporation conditions is it realistic to see D0 decreasing at lower altitudes? The fact that Nw is held constant should be an issue here, as one might expect Nw to strongly diminish lower in height, and maybe Do increase due to removal of the smaller drops evaporating? That is one illustration of my general comment 1. You need to show the vertical distribution, not a selected height, because it does not tell the full story.**

We now make a more detailed evaluation of the vertical profile for all cases. For Case 2 (Fig. 9) the forward-modelled mean Doppler velocity at both 94-GHz and 9.6-GHz are underestimated, but close to observations throughout the vertical profile, however the 9.6-GHz radar reflectivity is significantly underestimated in the lowest part of the profile, where evaporation is significant. This suggests D0 is well-constrained by the Doppler, and that the assumption of constant-Nw is leading to errors in the forward-modelled 9.6-GHz radar reflectivity.

In response to this and other questions about the assumption of constant-Nw, we have added Section 6, exploring the possibility of retrieving some additional information about the vertical structure: a retrieval of a linear profile of Nw is demonstrated for the warm rain case, in which we would expect collision–coalescence to lead to a decrease in drop number concentration toward the surface, combined with the increase in drop size.

For Case 2, if Nw is free to be represented by any linear profile (attached), we can improve the fit between forward-modelled 9.6-GHz radar reflectivity, with the retrieved Nw decreasing toward the surface and D0 increasing somewhat as the smallest drops evaporate, as you have suggested. This comes at the cost of an overestimate of 9.6-GHz mean Doppler velocity; this may indicate that the assumption of a constant shape factor mu=5 does not hold as the DSD becomes more monodisperse.

**3. Page 17, lines 23-24: You say that in this study you have used measurements to investigate the prospects for improved global rain rate retrievals from spaceborne Doppler radar. However in order to study that more completely you could have chosen to simulate from these aircraft observations what EarthCARE would actually measure, and whether you would be able to get similar results between degraded spaceborne measurements and higher-resolution higher-quality airborne measurements. I feel this piece of work is missing to really make that claim. In that respect I would remove mention of "spaceborne" in the title of the paper, not to give the impression that you are actually improving satellite rain rates with this technique, as I don't think you showed that.**

This is fair enough; the title has been changed.

**Specific comments:**

**1. Page 1, line 11: separate "between" and "light".**

Done

**2. Page 3, line 10, "...unattenuated wavelength...". This is incorrect, there is still large attenuation at X-band. This needs to be rephrased and errors associated with potential X-band attenuation assessed.**

That's true. Attenuation in the X-band is included in the radar forward-model.

Changes:

The line in question now reads "...at a less attenuated wavelength...", and further details has been added in Section 2.4 to make it clear that both W- and X-band radar attenuation are accounted for in the forward-model.

**3. Page 3, line 15, "dual-radar" : do you mean dual-frequency radar?**

Yes I did; fixed.

**4. Section 2.3.4 title : What about 94 GHz attenuation due to graupel or hail ? Something needs to be said about that in this paper.**

We have focused on stratiform precipitation here, where Doppler is expected to be most useful.

We expect that for EarthCARE a pre-check will flag convective precipitation, where

graupel and hail are expected, and in which the radar is quickly attenuated.

Changes:

Title changed to "Stratiform precipitation melting layer"

We note that "Melting of graupel and hail, usually associated with convective precipitation, are not considered in this melting layer model."

**5. Page 8, line 5 : "assume multiple scattering effects are negligible...". You should probably explain why you think that is reasonable (very small beamwidth). A comment is also needed to explain that this would need to be done in a spaceborne application.**

In response to this and comments from other reviewers, in this section we now describe the assumption of negligible multiple scattering for the airborne data, as well as the expected effects of MS on satellite Doppler radar.

Changes:

This section now reads:

> Multiple scattering effects on radar and lidar backscatter can be estimated within CAPTIVATE using Hogan (2008). Radar reflectivity enhancement due to multiple scattering is especially relevant to spaceborne radar measurements at millimeter wavelengths (Battaglia et al 2005), and the effects on Doppler radar measurements are expected to include both enhanced spectral broadening and modified mean Doppler velocity (Battaglia et al 2011); however, with the narrower beam of the airborne radar used in this study we can assume multiple scattering effects are negligible (Battaglia et al. 2007).

**6. Page 8, line 24: "as 3 dBZ, and as 0.5ms−1 for mean Doppler ...". This seems large for reflectivity and too small for Doppler velocity. Vertical air motions can be 1-2 ms-1 in the lower troposphere easily in the clouds you are interested in ... That brings up a question you need to address (sorry...): how sensitive to this value are the results?**

The ZvPIA retrievals are relatively insensitive to these uncertainties, however the Zv and ZPIA retrievals, being under-constrained, are significantly more sensitive to the choice of values.

In response to this and other comments we have reproduced the retrievals using more relaxed values that better fit with the expected combined measurement and forward-model errors: 3 dB for Z, 1.0 m/s for mean Doppler velocity, and 0.5 dB for PIA.

The following discussion has been added to this section:

> We have found that the weighting of errors between radar reflectivity and PIA is quite important for the retrieved rain rate, and that if only instrument errors are included the retrieval is not sufficiently constrained by PIA. This is believed to be because attenuation affects all forward-modelled radar reflectivity measurements in the same way, leading to them having strong error correlations. Error correlations are not accounted for in the R matrix, since they are profile-dependent and difficult to estimate, which can lead to the radar reflectivity measurements being over-weighted in the retrieval. To overcome this, we take the common approach (e.g. Weston et al. 2014) of inflating the reflectivity errors (and in our case somewhat reducing the errors in PIA) to better balance the information coming from the reflectivity profile and from PIA.

**7. Page 8, line 27, " .. by liquid water...". What about melting ice, graupel, hail ?**

This was intended to focus specifically on the interpretation of profiles of apparent radar reflectivity in rain; however, it is true that we had not sufficiently addressed the attenuation due to other hydrometeors, and this is now discussed in more detail in Section 2.3.4 on the melting layer (in response to comment 4).

**8. Page 14, line 31-32: What about at lower height? I would expect that if the Nw assumption is not satisfied lower down in the evaporating area, then D0 should have more errors and then Z(9.6 GHz) would be less good. It would actually be more interesting to show the whole vertical distribution instead of extracting one height for all these plots to demonstrate if the Nw assumption does create discrepancies on the vertical distribution of Z9.6. Hence my general comments 1 and 2.**

Indeed, the errors in 9.6-GHz reflectivity are largest toward the bottom of the profile where evaporation is most significant. Figures 5, 9 and 12 showing the averaged vertical profile for each case have been added, allowing evaluation of the retrieval against the radar variables at all heights.

In light of this and other questions about the retrieval of constant-Nw, we have added a brief section (Section 6) to the paper, to investigate the retrieval of more a linear gradient of Nw, which improve the fit to 9.6-GHz observations.

**9. Section 4.3.1, title: I think you could go to 12:46 for your comparisons? Why did you stop at 12:45 ?**

This was a pretty arbitrary division based on cloud-top height; however, we agree 12:46 seems like a more suitable cutoff based on radar reflectivity and PIA. The time periods have been changed in the title and discussion, and the period 12:41–12:46 is the subject of the added Section 6 on the retrieved profile of Nw.

[Figure]

**Fig. 1.** Comparison of linear-Nw and constant-Nw retrievals over the evaporating cold rain regime (Case 2); a similar figure for Case 3 is now included in Section 6 of the manuscript.

---

## Author Comment (AC4) · 6 Jul 2017

The comment was uploaded in the form of a supplement:
https://www.atmos-chem-phys-discuss.net/acp-2017-280/acp-2017-280-AC4-supplement.pdf

---

## Author Comment (AC5) · 6 Jul 2017

**List of changes**

[revised manuscript text omitted]

**2.3.2 Cost function and minimization**

Here we present a general description of the CAPTIVATE retrieval; justifications for the settings used in study are made in later subsections. The retrieval is made for each profile by iterating to find a state vector that minimizes a cost function, given by

$$J = \frac{1}{2}\delta\mathbf{y}^{\mathrm{T}}\mathbf{R}^{-1}\delta\mathbf{y} + \frac{1}{2}\delta\mathbf{x}^{\mathrm{T}}\mathbf{B}^{-1}\delta\mathbf{x} + J_c(\mathbf{x}) \tag{1}$$

where $\delta\mathbf{y} = \mathbf{y} - \mathbf{y}^f$ is the difference between the observed ($\mathbf{y}$) and forward-modelled ($\mathbf{y^f}$) measurements; $\mathbf{R}$ is the error covariance matrix of $\delta\mathbf{y}$, the sum of the error covariance matrices of the observations and the forward model; $\delta\mathbf{x} = \mathbf{x} - \mathbf{x}^a$ is the difference between the state ($\mathbf{x}$) and its a priori estimate ($\mathbf{x}^a$), and $\mathbf{B}$ is the error covariance matrix of $\mathbf{x}^a$ in which the diagonal elements are the error variances of $\mathbf{x}$; and $J_c(\mathbf{x})$ provides the capability to apply flatness and smoothing constraints to reduce the effect of observational noise on the state vector (Twomey, 1977). Additionally, profiles of retrieved variables can be represented smoothly as a set of cubic spline basis functions (Hogan, 2007, and in Section 2.3.3). The minimization of the cost function is carried out by iterating on the state vector beginning from the priors, in the direction of the first and second derivatives of the cost function (the Levenberg-Marquadt method; Rodgers, 2000).

**2.3.3 Rain state variables**

The rain DSD is given by a normalized Gamma function, of the form

$$N(D) = N_w \frac{\Gamma(4)}{3.67^4} \frac{(3.67+\mu)^{4+\mu}}{\Gamma(4+\mu)} \left(\frac{D}{D_0}\right)^{\mu} \exp\left(\frac{-(3.67+\mu)D}{D_0}\right). \tag{2}$$

This formulation is a function of three independent, physically meaningful parameters for the shape $\mu$, median drop size $D_0$, and drop number concentration $N_w$ of the DSD (Testud et al., 2001; Illingworth and Blackman, 2002). The shape factor $\mu$ is of secondary importance to $D_0$ and $N_w$ in terms of the radar reflectivity (Testud et al., 2001), and is poorly constrained by observations (e.g. Moisseev and Chandrasekar, 2007). In this retrieval we use $\mu = 5$, a value derived from both radar and distrometer studies (Wilson et al., 1997; Illingworth and Blackman, 2002). This simplifies the DSD to a 2-parameter function

of $D_0$ and $N_w$. The uncertainty due to the assumption of fixed-$\mu$ DSD is estimated to be $\pm 15\%$ of the rain rate (Wilson et al., 1997), and is included in the uncertainty estimates of the retrieved quantities.

Our primary state variable is the rain rate,

$$R = \frac{\rho_w \pi}{6} \int\limits_0^\infty N(D) D^3 v(D) \, dD \quad [\mathrm{kg\,m^{-2}\,s^{-1}}], \tag{3}$$

5  from the third moment of the DSD where $\rho_w$ is the density of liquid water, $v(D)$ is the raindrop terminal velocity as a function of drop size from Beard (1976) corrected for air density through the vertical profile. Hereafter we scale $R$ by a factor of 3600 to express rain rate in units of $\mathrm{mm\,h^{-1}}$. For all retrievals a prior $R$ of $0.1\,\mathrm{mm\,h^{-1}}$ is used. While a prior $R$ is not strictly necessary, it is applied in combination with a large prior variance ($\sigma(\ln R) = 4.0$), such that the retrieved $R$ is relatively insensitive to the prior unless the retrieval is poorly constrained by observations. We note that this value for the prior variance implies that before

10  the measurements are taken we assume there is a $44\%$ chance of $R$ lying between $0.01$ and $1.0\,\mathrm{mm\,h^{-1}}$, and a $56\%$ chance $R$ is outside these limits.

The second state variable is $N_w$, so that one state variable is an integral over the DSD and the second is a parameter of the DSD. Additional state variables increase the degrees of freedom of the retrieval, and require more information from observational variables to constrain the retrieval. Therefore we retrieve a single value of $N_w$ for each profile, with the physical

15  interpretation of representing $N_w$ as constant with height, or as the vertically-averaged value. The representation of $N_w$ as constant with height is not expected to be borne out in cases where evaporation or collision-coalescence processes modify the drop number concentration through the vertical profile. We take as the prior $N_w$ the number concentration intercept of the Marshall and Palmer (1948) DSD, $8 \times 10^6\,\mathrm{m^{-4}}$.

When few observational variables are available, a single-parameter retrieval of $R$ can be made by assuming that $N_w$ is

20  constant and equal to its prior, reducing the degrees of freedom so that $R$ is a function of $D_0$ alone. This is called the "$R$-only" retrieval, and is similar to CloudSat rain rate retrievals in which $N_w$ is assumed constant everywhere. When additional observational variables are available, such as the mean Doppler velocity, there may be sufficient information to also retrieve $N_w$; this is called the $R$-$N_w$ retrieval.

We use the natural logarithms of $R$ and $N_w$ as the state variables, with the effect that the values remain positive

25  everywhere and that the algorithm converges in fewer iterations. While in moderate stratiform rain $R$ is often close to invariant with height (e.g. Matrosov, 2007), processes such as evaporation in the lower atmosphere and collision–coalescence in warm clouds will lead to significant variation with height in many contexts. $R$ is therefore represented as the coefficients of a cubic spline basis function with $n$ elements (Hogan, 2007); this has the effect of ensuring the vertical profile of $R$ is smoothly varying and continuous with height, and also reducing the number of terms in the state vector. Table 1 summarises the rain

30  state variables, their prior values and uncertainties, and physical representation in each vertical profile. For $R$-only retrievals the state vector $\mathbf{x}$ for a vertical profile  is given by

$$\mathbf{x} = \ln \left[ R_1 \cdots R_n \right]^\mathrm{T} \tag{4}$$

while for the $R$-$N_w$ retrieval the  state vector is

$$\mathbf{x} = \ln \begin{bmatrix} R_1 \cdots R_n & N_w \end{bmatrix}^{\mathrm{T}} \tag{5}$$

where $N_w$ is assumed constant with height in each profile.

**Table 1.** Rain state variables $\mathbf{x_i}$, their prior values $\mathbf{x}_i^a$ and uncertainties $\sigma\left(\mathbf{x}_i^a\right)$. The profile of rain rate $R$ is always retrieved in each profile, while the drop number concentration parameter $N_w$ may be either retrieved or assumed equal to the prior; the melting layer thickness scaling $X_m$ is not retrieved in this study.

| $\mathbf{x}_i$ | $\mathbf{x}_i^a$ | $\sigma\left(\mathbf{x}_i^a\right)$ | Vertical representation |
| --- | --- | --- | --- |
| $\ln R$ | $\ln\left(0.1\,\mathrm{mm\,h^{-1}}\right)$ | 4.0 | Retrieved as the coefficients of a cubic spline basis function, with a spacing of $300\,\mathrm{m}$. |
| $\ln N_w$ | $\ln\left(8\times10^6\,\mathrm{m^{-4}}\right)$ | 3.0 | Retrieved as constant with height ($R$-$N_w$ retrievals), or not retrieved ($R$-only). |
| $X_m$ | $1.0\,\mathrm{km}$ | 0.0 | Not retrieved in this study. |

**2.3.4  **Stratiform precipitation** melting layer**

We employ a simplified representation of the melting layer in stratiform precipitation by applying radar attenuation between the lowest pixel in each profile classified as ice, and the highest pixel classified as rain, provided the two pixels are contiguous. Melting of graupel and hail, usually associated with convective precipitation, are not considered in this melting layer model. Following Matrosov (2008), it is assumed that the two-way attenuation of the melting layer $A$ is proportional to the rain rate $R$ at the first pixel just below the melting layer, and the two-way path length $X_m$ through the melting layer, such that

$$A = k_m X_m R \quad [\mathrm{dB}] \tag{6}$$

where the melting layer extinction coefficient $k_m$ is $2.2\,\mathrm{dB\,km^{-1}\left(mm\,h^{-1}\right)^{-1}}$ at $94\,\mathrm{GHz}$ and $0.04\,\mathrm{dB\,km^{-1}\left(mm\,h^{-1}\right)^{-1}}$ at $9.6\,\mathrm{GHz}$. The estimated attenuation through the melting layer is based on a Marshall–Palmer DSD for the rain below the melting layer (Matrosov, 2008), and is not modified to match the retrieved DSD in the profile. The thickness of the melting layer, and therefore the total attenuation, may also depend on the local temperature profile: as sufficient information to retrieve the total melting-layer attenuation may be available from the PIA and the attenuation inferred from the radar reflectivity gradient, we include the variable $X_m$ in the retrieval to represent the effect of melting layer thickness on radar attenuation;

however, in this study  $X_m$ is held constant with a value of $1.0\,\mathrm{km}$, allowing us to capture the effect of this uncertainty on the retrieved variables and their errors, without retrieving $X_m$.

**2.3.5 Radar forward model**

For a given state vector we estimate the corresponding measurements made by each instrument by forward modelling the scattering behaviour between the sensor and each gate for $94\,\mathrm{GHz}$ and $9.6\,\mathrm{GHz}$ radars, accounting for the effects of atmospheric gases, aerosols and hydrometeors.

The radar reflectivity factor of rain is a function of the sixth moment of the DSD,

$$Z = \int_0^\infty N(D) D^6 \gamma_f(D)\, dD \quad [\mathrm{dB\,Z}], \tag{7}$$

where $\gamma_f$ is the Mie–Rayleigh backscatter ratio at the radar frequency $f$, and is required for both $94\,\mathrm{GHz}$ and $9.6\,\mathrm{GHz}$ radars to account for non-Rayleigh scattering . At $94\,\mathrm{GHz}$ the uncertainty of assuming raindrops are spherical Mie scatters is approximately $5\,\%$ in integrated backscatter for a gamma DSD with median drop size $D_0 = 1.5\,\mathrm{mm}$, when compared against estimates for oblate spheroids (e.g. Thurai et al., 2007; Zhang et al., 2001) using the T-matrix method (Mishchenko et al., 1996).

Scattering and attenuation effects are included in the radar forward-model, so that the forward-modelled estimate of the apparent radar reflectivity ($Z_a$) is directly comparable to observations. Attenuation due to atmospheric gases and the dielectric factor of water are calculated from atmospheric temperature and humidity profiles (Liebe, 1985). Multiple scattering effects on radar and lidar backscatter can be estimated within CAPTIVATE using Hogan (2008). Radar reflectivity enhancement due to multiple scattering is especially relevant to spaceborne radar measurements at millimeter wavelengths (Battaglia et al., 2005), and the effects on Doppler radar measurements are expected to include both enhanced spectral broadening and modified mean Doppler velocity (Battaglia and Tanelli, 2011); however, with the narrower beam of the airborne radar used in this study we can assume multiple scattering effects are negligible (Battaglia et al., 2007).

Radar attenuation due to hydrometeors is quantified at each gate by the extinction coefficient

$$k = \frac{\pi}{4} \int_0^\infty Q(D) N(D) D^2\, dD \quad [\mathrm{m}^{-1}] \tag{8}$$

where $Q(D)$ is the extinction efficiency calculated from Mie theory (Mie, 1908). As for radar reflectivity, the uncertainty in extinction due to assuming spherical drops is less than $2\,\%$ for DSD with $D_0$ of $1.5\,\mathrm{mm}$. The gradient of extinction can be related to the gradient of apparent radar reflectivity and used to estimate the rain rate as suggested by Matrosov (2007). A second approach to quantifying attenuation due to hydrometeors is to measure the two-way path-integrated attenuation

$$\mathrm{PIA} = 2\frac{10}{\ln 10} \int_0^\infty k\, dz \quad [\mathrm{dB}] \tag{9}$$

for each profile. PIA is estimated from the radar reflectivity at the ocean surface, and used as an observational measurement. Both approaches are implemented simultaneously, so that  whereas the gradient method of Matrosov (2007) is applied only at moderate to heavy rain rates, where it can be assumed that the gradient of apparent radar reflectivity is dominated by attenuation, within the CAPTIVATE variational scheme the gradient of

5   $R$ and $k$ can be estimated simultaneously from the profile of radar reflectivity and the PIA.

    Finally the mean Doppler velocity is the reflectivity-weighted mean drop fallspeed,

$$\overline{v}_D = \frac{-\int_0^\infty N(D)D^6 v(D)\gamma_f(D)\,dD}{\int_0^\infty N(D)D^6 \gamma_f(D)\,dD} \quad [\mathrm{m\,s}^{-1}] \tag{10}$$

where the terminal fallspeed of drops $v(D)$ is from the empirical formulation of Beard (1976) scaled to account for air density changes with altitude, and where positive velocities are toward the ground. The forward-modelled mean

10   Doppler velocity is calculated assuming zero vertical air motion; therefore the difference between the forward-modelled and observed mean Doppler velocities will include a contribution from the vertical air motion, which is treated as an observational uncertainty.

    The observed variables, their observational uncertainties and their vertical representation are summarized in Table 2. The uncertainties in the observational variables include both

15   the specified measurement errors for the CRS instrument (Li et al., 2004) and the estimated uncertainties in the radar forward model. We have found that the weighting of errors between radar reflectivity and PIA is quite important for the retrieved rain rate, and that if only instrument errors are included the retrieval is not sufficiently constrained by PIA. This is believed to be because attenuation affects all forward-modelled radar reflectivity measurements in the same way, leading to them having

20   strong error correlations. Error correlations are not accounted for in the $\mathbf{R}$ matrix, since they are profile-dependent and difficult to estimate, which can lead to the radar reflectivity measurements being over-weighted in the retrieval. To overcome this, we take the common approach (e.g. Weston et al., 2014) of inflating the reflectivity errors (and in our case somewhat reducing the errors in PIA) to better balance the information coming from the reflectivity profile and from PIA.

**Table 2.** Observational variables $\mathbf{y_i}$ for Doppler radar, and their estimated uncertainties $\sigma(\mathbf{y_i})$ as used in the retrieval. Apparent radar reflectivity $Z_a$ and mean Doppler velocity $\overline{v}_D$ are measured at each gate, while PIA is estimated from the radar reflectivity over the ocean surface.

| $\mathbf{y_i}$ | $\sigma(\mathbf{y_i})$ | Vertical representation |
|---|---|---|
| $Z_a$ | 3.0 dB | At each radar gate |
| $\overline{v}_D$ | 1.0 m s[-1] | At each radar gate |
| PIA | 0.5 dB | Integrated for each profile |

[revised manuscript text omitted]
 retrieval is evaluated by forward-modelling all $94\,\mathrm{GHz}$ and $9.6\,\mathrm{GHz}$ radar variables, whether or not they were assimilated in the retrieval, and comparing against the observations.

**4.1 Case 1: moderate rain from melting ice, 22 July 2007**

Stratiform rain from melting ice provides a test of many of the simplifying assumptions made in rain retrievals. At moderate and heavy rain rates we expect $R$ to be close to constant with height, unless significant evaporation is evident (Haynes et al., 2009). $N_w$ may be expected to be close to values deemed typical by Marshall and Palmer (1948) or Testud et al. (2001), i.e. between $2.0 \times 10^6 - 8.0 \times 10^6\,\mathrm{m}^{-4}$, and constant with height (Tokay and Short, 1996). From in situ measurements of stratiform rain we expect median drop sizes  to be in the range $1.0 - 1.5\,\mathrm{mm}$ (Tokay and Short, 1996).

Between 15:54 and 16:03 UTC on 22 July 2007 ER-2 overflew approximately $110\,\mathrm{km}$ of precipitating stratiform cloud  around $50\,\mathrm{km}$ south of the coast of Panama (Fig. 2). Radar, lidar and radiometer measurements (Fig. 3) reveal distinct regimes of light, moderate and heavy rain below a melting layer around $4.5\,\mathrm{km}$ above sea level, contiguous with ice clouds with tops between $6 - 10\,\mathrm{km}$. The scene is overlain by cirrus between $10 - 15\,\mathrm{km}$, which is primarily detected by the lidar. In light rain between 15:54 and 15:55 UTC the $94\,\mathrm{GHz}$ radar is barely attenuated. Moderate stratiform rain follows from 15:55 and 16:03 UTC, with a strong $9.6\,\mathrm{GHz}$ bright band evident, and $94\,\mathrm{GHz}$ PIA between 5 and $20\,\mathrm{dB}$. Finally a heavy shower is embedded within the moderate rain between 16:01 and 16:02 UTC. In the latter regime the $94\,\mathrm{GHz}$ radar is completely attenuated such that PIA saturates around $60\,\mathrm{dB}$ ; $94\,\mathrm{GHz}$ radar reflectivity and mean Doppler velocity measurements are therefore not available within these  heaviest rain profiles.

The retrieved variables (Figs. 4a–4e), and forward-modelled $94\,\mathrm{GHz}$ and $9.6\,\mathrm{GHz}$ radar measurements (Figs. 4f–4j) are compared for the ZvPIA, Zv and ZPIA retrievals. We evaluate the retrievals at a height of $3\,\mathrm{km}$ above sea level, approximately $1\,\mathrm{km}$ below the melting layer.

**4.1.1 Moderate rain (15:55–16:01 and 16:02–16:03 UTC)**

In the moderate rain regime the ZvPIA retrieval estimates rain rates of $1.0 - 2.0\,\mathrm{mm\,h}^{-1}$ at the melting layer. In profiles with strong attenuation (PIA up to $20\,\mathrm{dB}$), $R$ is close to constant from the melting layer to the surface; conversely, in less attenuated profiles (with PIA around $10\,\mathrm{dB}$) some evaporation is evident, with $R$ reducing to $0.1 - 1.0\,\mathrm{mm\,h}^{-1}$ at the surface (Fig. 4a). Estimates of $N_w$ are consistently between $10^6$–$10^7\,\mathrm{m}^{-4}$ in this regime (Fig. 4c), close to the Marshall and Palmer (1948) value, while. $D_0$ is around $1.0\,\mathrm{mm}$ at the melting layer and decreases,  somewhat toward the surface in profiles where evaporation is strong (Fig. 4d). Forward-modelled $94\,\mathrm{GHz}$ radar measurements agree with observations at $3\,\mathrm{km}$ (Figs. 4f–h), as expected since the retrieval minimizes differences between the observed and forward-modelled variables.  $9.6\,\mathrm{GHz}$ radar measurements  forward-modelled from the retrieved state show generally good agreement with independent observations at $3\,\mathrm{km}$ above sea level (Figs. 4i and 4j), although $9.6\,\mathrm{GHz}$ radar reflectivity is overestimated by as much as $3\,\mathrm{dB}$ in profiles with strong evaporation between 15:58 and 16:00 UTC, and mean Doppler velocity is underestimated in the profiles with the heaviest rain.

The averaged vertical profiles of the ZvPIA retrieval in moderate rain (Fig. 5), show that the forward-modelled $94\,\mathrm{GHz}$ radar reflectivity is over-estimated near the surface, while the largest error in $9.6\,\mathrm{GHz}$ is in the mean Doppler velocity in the lowest $2-3\,\mathrm{km}$. We suggest that these errors in the forward-modelled variables through the vertical profile relate to the representation of $N_w$ as constant with height, such that the effects of evaporation on the DSD—a decrease in concentration of the smallest drops—is not resolved. In a vertically-averaged sense, however, the ZvPIA retrieval is broadly able to reproduce the $9.6\,\mathrm{GHz}$ radar reflectivity, while slightly under-estimating mean Doppler velocity.

 The ZPIA and Zv retrievals  illustrate  the contributions of mean Doppler velocity and PIA to a ZvPIA  retrieval, and the ambiguities that arise in under-constrained retrievals. Both ZPIA and Zv retrievals are considerably more sensitive to the selection of priors and uncertainties than the ZvPIA retrieval. At $3\,\mathrm{km}$ above sea level (Fig. 4), ZPIA estimates of $R$ in the moderate rain regime are close to those of ZvPIA, but $N_w$ and $D_0$ differ significantly, with ZPIA estimating a much higher concentration of smaller drops than ZvPIA. Forward-modelled mean Doppler velocity shows that this retrieval leads to large  errors in  drop fallspeeds. The Zv retrieval tends to underestimate  the rain rate in this regime by up to an order of magnitude, tending toward the prior $R$ of $0.1\,\mathrm{mm\,h^{-1}}$, with the exception of the strongly attenuated profiles 15:57–15:58 UTC where Zv is able to reproduce the observed PIA from the profiles of radar reflectivity and mean Doppler velocity: while $D_0$ is well constrained by the mean Doppler velocity, without a constraint on PIA  the retrieved drop number concentration and rain rate are  lower than that estimated from ZvPIA; the forward-modelled observations confirm  that the Zv retrieval meets reflectivity and mean Doppler velocity constraints, but tends to  represent  weakly-attenuating profiles of rain; the forward-modelled $9.6\,\mathrm{GHz}$ variables  show that this retrieval leads to a significantly under-estimate  radar reflectivity at this frequency .

[revised manuscript text omitted]

Without a PIA constraint, Zv retrievals in the moderate rain regime tend toward weakly attenuated profiles with $N_w$ less than $10^6\,\mathrm{m}^{-4}$, where $R$ is close to the prior. This leads to underestimates of radar reflectivity by more than $10\,\mathrm{dB}$; the mean Doppler velocity is reasonably well-constrained, but broadly underestimated by around $1\,\mathrm{m\,s}^{-1}$. A secondary mode with $N_w$ close to the prior and $R$ greater than $1\,\mathrm{mm\,h}^{-1}$ represents the strongly attenuated profiles of moderate rain in which Zv comes close to reproducing the observed PIA. Light rain profiles are represented with $N_w \approx 10^6\,\mathrm{m}^{-4}$, somewhat overestimating mean Doppler velocity.

ZvPIA resolves distinct modes for light and moderate rain regimes in the retrieved variables, and each mode corresponds well to the observed $9.6\,\mathrm{GHz}$ radar measurements: the moderate rain regime is represented with heavier rain than the Zv retrieval, but with a lower concentration of smaller drops than the ZPIA retrieval; the light rain regime is similar to that of the Zv retrieval, where the negligible PIA provides little additional information. Both rain regimes have $N_w$ around $10^6\,\mathrm{m}^{-4}$, consistent with the average value of $2 \times 10^6\,\mathrm{m}^{-4}$ for stratiform rain found by Testud et al. (2001). The $9.6\,\mathrm{GHz}$ radar reflectivity is well-represented across both rain regimes, however the mean Doppler velocity shows that drop fallspeed is slightly under-estimated in moderate rain, and over-estimated in the light rain; this may be due to the retrieval assuming $N_w$ is constant with height in each profile, such that any variations in the DSD with height are expressed as changes in drop size, rather than in drop number concentration.

We have retrieved $R$ as a function of both $D_0$ and $N_w$ for a case of stratiform rain from melting ice, including  rain rates  from light rain as low as $10^{-3}\,\mathrm{mm\,h}^{-1}$, to a heavy shower with $R$ up to $10\,\mathrm{mm\,h}^{-1}$. The retrieved $N_w$ was around $10^6\,\mathrm{m}^{-4}$ throughout the case, which is consistent with expectations for average drop number concentrations in this context; the exception is in the heavy rain shower where the $94\,\mathrm{GHz}$ radar becomes fully attenuated, and insufficient information is available for a $R$-$N_w$ retrieval. The Zv retrieval, an analogue for Doppler radar retrievals over land, performed very well in light rain where total attenuation is close to zero, but tended towards the $R$ prior in profiles of moderate where attenuation makes radar reflectivity ambiguous, and $R$ tends to be under-estimated. ZPIA retrievals without mean Doppler velocity information tend to estimate $R$ broadly accurately, but retrieve DSDs with a high  concentration of small drops, leading to errors with respect to the independent radar measurements; indeed, since the estimated $N_w$ were close to expectations in this context, a good non-Doppler retrieval of $R$ could have been made by assuming the value of $N_w$ is equal to the prior.

**4.2 Case 2: evaporating rain from melting ice, 22 July 2007**

We now evaluate the $R$-$N_w$ retrieval for a case of very light rain from melting ice much of which evaporates before reaching the ground. ER-2 overflew a $60\,\mathrm{km}$ section of stratiform cloud $300\,\mathrm{km}$ south of Costa Rica, between 13:12–13:17 UTC on 22 July 2007. Light rain was observed below  clouds with tops between $10-12\,\mathrm{km}$ (Fig. 7). Below the melting layer both $94\,\mathrm{GHz}$ and $9.6\,\mathrm{GHz}$ radar reflectivity are less than $10\,\mathrm{dBZ}$ and decrease toward the surface; the exception is a region of higher $9.6\,\mathrm{GHz}$ radar reflectivity between 13:16–13:17 UTC,  $94\,\mathrm{GHz}$

[revised manuscript text omitted]

**4.3.1 Moderate rain (12:41–12:45 UTC)**

The ZvPIA retrieval resolves a strong increase in rain rate from cloud-top, where $R$ is between $0.1-1.0\,\mathrm{mm\,h^{-1}}$, to the surface, where $R$ increases to $1.0-10.0\,\mathrm{mm\,h^{-1}}$. Retrieved $N_w$ is consistently around $10^{10}\,\mathrm{m^{-4}}$ in the moderate rain regime, several orders of magnitude greater than were estimated for rain from melting ice; accordingly, the drops are much smaller, with $D_0$ increasing from $0.1-0.3\,\mathrm{mm}$ at cloud-top to $0.2-0.5\,\mathrm{mm}$ near the surface. At $1\,\mathrm{km}$ above sea level the $94\,\mathrm{GHz}$ radar measurements correspond very well to the forward-modelled variables. The $9.6\,\mathrm{GHz}$ radar reflectivity is also  close to the forward-model; however, while the forward-modelled mean Doppler velocity at $9.6\,\mathrm{GHz}$ also tracks well with observations, peaks associated with the heaviest precipitation features are not resolved.

The vertical structure of $94\,\mathrm{GHz}$ and $9.6\,\mathrm{GHz}$ radar reflectivity is well represented in the ZvPIA retrieval over the moderate warm rain regime (Fig 12); however, mean Doppler velocity is under-estimated by around $1\,\mathrm{m\,s^{-1}}$ in the lowest $1\,\mathrm{km}$ at both radar frequencies. The retrieval of constant-$N_w$ for each profile allows a broadly satisfactory retrieval of the rain DSD with a good fit to observations, but evaluation of the full vertical profiles shows that some microphysical process is not resolved: in warm rain we expect collision and coalescence to lead to both an increase in drop size and a decrease in drop number concentration toward the surface. It seems likely, as for the representation of evaporation in case 2, that while the retrieval of $N_w$ allows for an improved retrieval of the DSD across a range of rain regimes, there are limits to the vertical variability of the DSD that can be resolved with a height-invariant $N_w$.

The ZPIA retrieval closely resembles ZvPIA; this includes matching estimates of $D_0$, despite having no constraint on drop size from mean Doppler velocity.  Zv retrieves similar $D_0$, but frequently under-estimates $N_w$ by as much as 2 orders of magnitude: the Zv-retrieved DSD has fewer drops and negligible PIA at $94\,\mathrm{GHz}$, which corresponds to very large errors in forward-modelled $9.6\,\mathrm{GHz}$ radar reflectivity. Unlike the stratiform rain cases, here PIA is more important for an accurate retrieval than mean Doppler velocity: the mean Doppler velocity may be less sensitive to the changes in terminal fallspeed due to variations in the sizes of small drops, while PIA in combination

with radar reflectivity provide an effective constraint on the number concentration because only a DSD with many small drops satisfies the observed strong attenuation and low radar reflectivity.

**4.3.2 Light rain (12:46–12:51 UTC)**

In the light warm rain ZvPIA estimates patchy precipitation features with $R$ between $0.01 - 0.5\,\mathrm{mm\,h^{-1}}$ and median drop sizes around $0.1 - 0.3\,\mathrm{mm\,h^{-1}}$, similar to values at the tops of the deeper warm clouds, but without significant drop growth toward the surface. The retrieved $N_w$ in the lightest rain profiles is around $10^8 - 10^9\,\mathrm{m^{-4}}$, but returns to $10^{10}\,\mathrm{m^{-4}}$ where heavier rain features are evident. The forward-modelled radar reflectivities are close to observations at $1\,\mathrm{km}$, while the mean Doppler velocity again matches the lower range of measurements, but not the  peaks.  ZPIA estimates $R$ similar to ZvPIA in the stratus regime, but without a Doppler velocity constraint in the lightest profiles, ZPIA retrieves fewer, larger drops, with $N_w$ tending toward the prior at $8 \times 10^6\,\mathrm{m^{-4}}$. In contrast, Zv is very similar to ZvPIA in the light .

In warm rain, we have retrieved $N_w$ several orders of magnitude greater than the Marshall–Palmer value, with $D_0$ in the range $0.1 - 0.5\,\mathrm{mm}$ in rain rates from very light drizzle up to $10\,\mathrm{mm\,h^{-1}}$ in the heaviest profiles . The contribution of PIA and mean Doppler velocity to $R$-$N_w$ retrievals in warm rain differs from that in rain from melting ice: while Doppler is required to retrieve $N_w$ when attenuation is low, it is possible to retrieve $N_w$ without Doppler in strongly attenuated profiles of warm cloud, where the combination of low radar reflectivity and high attenuation can only be due to a high concentration of small drops.

**5 Dual-frequency radar retrievals**

ER-2 aircraft measurements from TC4 provide a rare opportunity for airborne observations with multiple Doppler radars. In this study we have primarily used the $9.6\,\mathrm{GHz}$ radar to evaluate retrievals made with the $94\,\mathrm{GHz}$ radar; however, we can also use the dual-frequency radar measurements to exploit the different scattering behaviours and retrieve additional information about the DSD. Dual-frequency ratio (DFR) and differential Doppler velocity (DDV) techniques have been applied to retrievals from ER-2 measurements during the CRYSTAL-FACE field experiment over Florida in 2002 (Liao et al., 2008, 2009), and Tian et al. (2007) exploited dual-frequency Doppler radar to retrieve rain DSD and vertical air motion for light stratiform rain from the same experiment. The CAPTIVATE framework can combine information from two radars, resolving differential non-Rayleigh scattering and mean Doppler velocities from multiple wavelengths.

We compare the dual-frequency radar retrievals, with and without mean Doppler velocity measurements against the ZvPIA $94\,\mathrm{GHz}$ retrieval for Case 1 , which covered a wide range of rain intensities, including a region in which the $94\,\mathrm{GHz}$ radar was fully attenuated (Fig. 13). The dual-frequency radar retrieval estimates of $R$ are consistent with those from $94\,\mathrm{GHz}$ , with the exception of the non-Doppler dual-frequency radar retrieval in  light rain, where

a high concentration of small drops is estimated, leading to an over-estimate of $R$; in much of the lightest rain  the hydrometeors may be below the sensitivity of the $9.6\,\mathrm{GHz}$ instrument, so that the dual-frequency radar retrieval tends toward that  from the $94\,\mathrm{GHz}$ radar. In the heavy shower, where the ZvPIA estimates of $Z$ have large uncertainties and $N_w$ is very poorly constrained due to complete extinction of the $94\,\mathrm{GHz}$ radar beam, the dual-frequency radar retrievals use  $9.6\,\mathrm{GHz}$ measurements alone to estimate $R$ around $10\,\mathrm{mm\,h^{-1}}$; $N_w$ remains in the range $10^6 - 10^7\,\mathrm{m^{-4}}$ as in the surrounding moderate rain, and $D_0$ is estimated between $1 - 2\,\mathrm{mm}$. This illustrates that the $94\,\mathrm{GHz}$ retrieval was capable of a cautious estimate of $R$ with large retrieval uncertainty, based on the gradient of radar reflectivity and saturated PIA; however,  estimates of $N_w$ cannot be justified when the radar is fully attenuated. The greatest errors in the non-Doppler dual-frequency radar retrieval are in forward-modelled Doppler velocity for the evaporating moderate rain profiles between 15:58–16:00 UTC, where a higher concentration of smaller drops is retrieved; in this circumstance the addition of Doppler velocity information leads to a stronger retrieval than a second radar wavelength. Overall the close agreement of the $94\,\mathrm{GHz}$ Doppler radar retrievals with the dual-frequency Doppler retrieval is a promising result, indicating that a single frequency Doppler radar is sufficient for a retrieval of $R$ and $N_w$ within the limits of radar attenuation.

**6 Retrieving vertical profiles of $N_w$**

We have demonstrated the retrieval of rain rate as a function of both $D_0$ and $N_w$, making the simplifying assumption that $N_w$ is constant with height in each profile. We argue that this is a significant improvement over retrievals in which $N_w$ is assumed constant everywhere, and have retrieved values of $N_w$ ranging over more than five orders of magnitude between light rain from melting ice and warm rain from liquid clouds; however, evaluation against $9.6\,\mathrm{GHz}$ radar measurements showed that features within the vertical profile are not always accurately resolved, with the most significant errors near the surface in cases where microphysical processes are expected to affect the DSD with height.

It is of interest to represent changes in $N_w$ through the vertical profile; however, there are limits to the degrees of freedom that can be retrieved with the available observed variables. In this section we explore the potential for one additional degree of freedom, by allowing each profile of $N_w$ to be represented by a linear gradient, as  explored in Rose and Chandrasekar (2006) for a dual-frequency retrieval. Here the state vector becomes:

$$\mathbf{x} = \ln\left[R_1 \cdots R_n \quad \overline{N_w} \quad N_w' \right]^\mathrm{T} \tag{12}$$

where $\overline{N_w}$ is the average $N_w$ through the profile and $N_w'$ is the gradient with height.

A retrieval in which $N_w$ is represented by a linear profile ("linear-$N_w$") is compared against the "constant-$N_w$" ZvPIA retrieval, using the average profiles of retrieved and forward-modelled variables for a ZvPIA retrieval of moderate warm rain from Case 3 (Fig. 14). The linear-$N_w$ retrieval significantly improves the fit with $94\,\mathrm{GHz}$ observed variables below $1.5\,\mathrm{km}$, where the constant-$N_w$ retrieval underestimates mean Doppler velocity and overestimates radar reflectivity. The linear-$N_w$ retrieval is also better able to forward-model the independent $9.6\,\mathrm{GHz}$ radar variables, with near-surface errors in mean Doppler velocity significantly reduced. The linear-$N_w$ retrieval resolves a gradient in $N_w$ from around $10^{11}\,\mathrm{m^{-4}}$ at cloud-top to $10^9\,\mathrm{m^{-4}}$

near the surface, and a steeper gradient of $D_0$, with corresponding drop size increasing from almost $0.1\,\mathrm{mm}$ near cloud-tops to around $0.5\,\mathrm{mm}$ at the surface. The changes in $D_0$ and $N_w$ through the vertical profile have relatively minor effects on the profile-averaged rain rate, with the retrieved $R$ increasing somewhat above $2\,\mathrm{km}$, and a decreasing below $0.5\,\mathrm{km}$ by around a factor of 2.

With an additional degree of freedom, the linear-$N_w$ retrieval from $94\,\mathrm{GHz}$ radar exhibited increased temporal variability between profiles of the retrieved variables; however, a dual-frequency retrieval (as in Section 5) including a linear representation of $N_w$ estimated substantially similar profiles of $N_w$ and $D_0$ for this case, indicating that the retrieved profiles are robust when better constrained by additional observations. The retrieval of a linear gradient $N_w$ leads to an improved representation of the vertical profile, both as evaluated against independent $9.6\,\mathrm{GHz}$ radar variables, and in that the retrieved profiles of $N_w$ and $D_0$ qualitatively meet expectations for collision and coalescence processes in warm rain. Nevertheless, we note that the profile-averaged $N_w$ is close to that retrieved with a height-invariant $N_w$, and that the corresponding changes in retrieved $R$ are small in a vertically-averaged sense.

**7  Discussion and conclusions**

The upcoming ESA/JAXA EarthCARE satellite will include the $94\,\mathrm{GHz}$ cloud profiling radar, the first Doppler radar in space. In this study we have used airborne Doppler radar measurements to investigate the prospects for improved  rain retrievals from a spaceborne Doppler radar, with a focus on improving upon $94\,\mathrm{GHz}$ radar rain retrievals from CloudSat in two key respects: (1) to facilitate rain rate estimates over land, and; (2) to reduce uncertainties in rain rate estimates by retrieving an additional parameter of the raindrop size distribution (DSD). Retrievals over a range of stratiform rain regimes were made using measurements from the $94\,\mathrm{GHz}$ Doppler radar aboard the ER-2 aircraft during the TC4 field campaign over the tropical Pacific in 2007, and evaluated against independent measurements from a less attenuated $9.6\,\mathrm{GHz}$ Doppler radar.

The CAPTIVATE algorithm has been developed for rain, cloud and aerosols retrievals from the synergy of active and passive instruments; within the variational scheme, multiple observational variables can be combined as available, and the retrieved variables and their physical representation can be configured at runtime. It is therefore possible with CAPTIVATE to combine the information from multiple radar measurements, and to estimate uncertainties in retrieved variables propagated from uncertainties in the observations and forward-models.

We have shown that the ambiguities of rain rate retrievals at strongly attenuated radar frequencies can be resolved by either an estimate of path-integrated attenuation (PIA) or by the profile of mean Doppler velocity, a measure of drop terminal fallspeed relating to drop size, and which is not affected by partial attenuation of the radar beam. The latter has potential applications to making estimates of rain rate over land , where PIA is more difficult to estimate from the surface backscatter. Furthermore, information from both PIA and mean Doppler velocity can be used to retrieve the rain rate as a function of both median drop size $D_0$ and drop number concentration $N_w$, improving upon significant uncertainties in rain rate estimates owing to assumptions about the DSD.

Retrievals of rain rate $R$ and drop number concentration $N_w$ using combined radar reflectivity, PIA and mean Doppler velocity measurements from 94 GHz radar were evaluated over three cases of tropical stratiform rain over the ocean. The selected cases covered a range of rain rates from virga to heavy showers from melting ice and liquid clouds. The 94 GHz radar was fully attenuated in profiles with rain rates around $10\,\mathrm{mm\,h^{-1}}$  below a melting layer above 4 km, and in rain from liquid clouds with tops around 3 km. The attenuation of the 94 GHz radar places an upper limit on the rain profiles that can be retrieved; however, in the mid-latitudes where the melting layer is lower, it may be possible to make retrieval up to higher rain rates before the radar is fully attenuated. The 9.6 GHz radar wavelength was subsequently used to make a dual-frequency radar retrieval (Section 5). In profiles where the 94 GHz was fully attenuated, the dual-frequency estimates of rain rate were consistent with those derived from the gradient of 94 GHz radar reflectivity. Retrieved values of $N_w$ ranged from $10^5\,\mathrm{m^{-4}}$ in light rain from melting ice, with $D_0$ around $1.0-1.5\,\mathrm{mm}$, up to $N_w$ of $10^{10}\,\mathrm{m^{-4}}$ in moderate rain from liquid cloud, where $D_0$ was around $0.1-0.3\,\mathrm{mm}$.

In many contexts collision–coalescence, evaporation and breakup processes are expected to modify the DSD through the vertical profile. With $N_w$ retrieved as constant with height the retrieval was broadly able to represent the major features of the vertical profile when evaluated against independent 9.6 GHz radar measurements, but errors in the gradient of mean Doppler velocity suggested that the effects of evaporation or collision–coalescence on the DSD through the profile were not resolved. We demonstrated that the 94 GHz radar measurements, including both mean Doppler velocity and PIA, are sufficient to retrieve a more complex representation of $N_w$ with height (Section 6): in warm rain from liquid cloud it was possible to resolve the decrease in drop concentration and increase in drop size toward the surface consistent with collision–coalescence, while improving on errors with respect to forward-modelled 9.6 GHz radar measurements. The retrieval of a vertical gradient of $N_w$ has potential applications to both single- and multiple-frequency retrievals of precipitation (e.g. Rose and Chandrasekar, 2006), but must be constrained by sufficient observational variables.

Furthermore, in combination with PIA, the Doppler velocity information provided sufficient information to make robust retrievals of $R$-$N_w$ across a range of rain regimes. In light rain with negligible PIA, Doppler velocity is sufficient to retrieve $R$ and $N_w$, suggesting the possibility of using Doppler radar for $R$-$N_w$ retrievals of light rain over land without PIA; however, in moderate rain rates PIA is necessary to constrain the retrieval. Satisfactory retrievals of rain rate may be made over land by assuming $N_w$ constant, especially for stratiform precipitation, or PIA could be estimated from the land surface backscatter with a large observational uncertainty (as in Iguchi et al., 2009), which may provide sufficient information to resolve the ambiguity between weakly and strongly attenuating profiles. A robust method of using Doppler radar to estimating rain rate over land will be the subject of future work.

With a constraint on PIA, the gradient of apparent radar reflectivity can be used to estimate $R$. While Doppler velocity is generally required to retrieve $N_w$, in moderate warm rain from liquid clouds the combination of low radar reflectivity and strong attenuation was sufficient to retrieve the high concentration of small drops typical of warm rain, without the need for Doppler

velocity information: this finding may be applicable to retrievals of the drop number concentration in warm rain observed by CloudSat.

We have demonstrated the contribution of mean Doppler velocity to assimilating drop size information in estimates of rain rate. With the first Doppler radar in space, EarthCARE synergy retrievals will exploit novel measurements to improve the  satellite remote sensing of clouds and precipitation.  Future work will focus on understanding the application of these retrievals to spaceborne Doppler radar, including the effects of multiple scattering and non-uniform beam filling on the Doppler measurements. As part of EarthCARE radar–lidar–radiometer synergy retrievals, improved global estimates of rain rate and drop size  will provide new insights into the interactions of clouds and precipitation through the atmospheric profile.

*Acknowledgements.* This work was supported by the National Centre for Earth Observation (NCEO) and European Space Agency Grant 4000112030/15/NL/CT, with computing resources provided by the University of Reading. Dr. Tian's research is supported by NASA Precipitation Measurement Mission and Remote Sensing Theory. We thank Gerry Heymsfield and the ER-2 radar engineers for collecting CRS and EDOP data, Dennis Hlavka (NASA-GSFC) for assistance with CPL data, and Stephen Platnick and Howard Tan (NASA-JPL) with MAS/MASTER radiometer data. ERA-Interim data are produced and distributed by ECMWF, and hosted by the Centre for Environmental Data Analysis.

We are grateful to Alain Protat and two anonymous referees for their constructive feedback, and Ross Bannister, Nancy Nichols, Lars Isaksen, Elias Holm and Mike Rennie for helpful discussions.

[revised manuscript text omitted]

**Figure 12.** Averaged profiles of moderate rain between 12:41 and 12:46 UTC on 29 July 2007. Forward-modelled $94\,\text{GHz}$ radar reflectivity (a), mean Doppler velocity (b) and PIA (c); forward-modelled $10\,\text{GHz}$ radar reflectivity (d) and mean Doppler velocity (e); and retrieved rain rate (f), median drop size (g), number concentration parameter (h) for ZPIA, Zv and ZvPIA retrievals. The number of profiles included at each height is indicated in (i). Shading and dashed lines indicates the $1\text{-}\sigma$ standard deviation of the retrieved and derived variables.

[Figure]

**Figure 13.** Time series of dual-frequency (DF) retrievals with and without Doppler, compared against the $94\,\mathrm{GHz}$ ZvPIA retrieval for Case 1 on 22 July 2007. Retrieved state and derived variables (left), and forward-modelled radar measurements (right) for the three retrievals are shown at a height of $3.0\,\mathrm{km}$ above sea level (indicated with a light dashed line in the left-hand scenes); while the full scenes of $R$ (a) and $D_0$ (d) are shown for the dual-frequency Doppler retrieval. Shading indicates the $1\text{-}\sigma$ uncertainty in the retrieved and derived variables. Dark dashed lines (right) indicate the observed radar measurements.

[Figure]

**Figure 14.** Averaged profiles of moderate rain between 12:41 and 12:46 UTC on 29 July 2007. Forward-modelled $94\,\mathrm{GHz}$ radar reflectivity (a), mean Doppler velocity (b) and PIA (c); forward-modelled $10\,\mathrm{GHz}$ radar reflectivity (d) and mean Doppler velocity (e); and retrieved rain rate (f), median drop size (g), drop number concentration parameter (h) for ZvPIA retrievals in which $N_w$ is represented as a constant with height, and as a linear gradient. The number of profiles included at each height is indicated in (i). Shading and dashed lines indicate the $1$-$\sigma$ standard deviation of the retrieved and derived variables.